# Mix Data or Merge Models? Balancing the Helpfulness, Honesty, and Harmlessness of Large Language Model via Model Merging

**Jinluan Yang**[1], **Dingnan Jin**[2], **Anke Tang, Li Shen, Didi Zhu**[1], **Zhengyu Chen**[1]
**Ziyu Zhao**[1], **Daixin Wang**[2] **Qing Cui**[2], **Zhiqiang Zhang**[2], **Jun Zhou**[2]
**Fei Wu**[1], **Kun Kuang**[1]*
[1] Zhejiang University; [2] Ant Group
yangjinluan@zju.edu.cn,

## Abstract

Achieving balanced alignment of large language models (LLMs) in terms of Helpfulness, Honesty, and Harmlessness (3H optimization) constitutes a cornerstone of responsible AI. Existing methods like data mixture strategies face limitations, including heavy reliance on expert knowledge and conflicting optimization signals. While model merging offers parameter-level conflict-resolution strategies through integrating specialized models' parameters, its potential for 3H optimization remains underexplored. This paper systematically compares the effectiveness of model merging and data mixture methods in constructing 3H-aligned LLMs for the first time, revealing previously overlooked collaborative and conflict relationships among the 3H dimensions and discussing the advantages and drawbacks of data mixture (*data-level*) and model merging (*parameter-level*) methods in mitigating the conflict for balanced 3H optimization. Specially, we propose a novel **R**eweighting **E**nhanced task **S**ingular **M**erging method, **RESM**, through outlier weighting and sparsity-aware rank selection strategies to address the challenges of preference noise accumulation and layer sparsity adaptation inherent in 3H-aligned LLM merging. Extensive evaluations can verify the effectiveness and robustness of RESM compared to previous data mixture (2%-5% gain) and model merging (1%-3% gain) methods in achieving balanced LLM alignment.

## 1 Introduction

Large language models (LLMs) have demonstrated remarkable capabilities across diverse natural language processing tasks [1–3]. However, their reliable deployment necessitates balanced optimization across three critical dimensions: *Helpfulness* (providing accurate and task-aligned responses), *Honesty* (avoiding hallucinations and misinformation), and *Harmlessness* (preventing toxic or unethical outputs), collectively termed *3H optimization* [4–7]. While recent alignment techniques such as constitutional AI [8], reinforcement learning from human feedback (RLHF) [9], and Direct Preference Optimization (DPO) [10] have improved individual aspects of 3H, seeking a balance remains a significant challenge. For instance, models optimized for helpfulness may inadvertently generate harmful content [11],

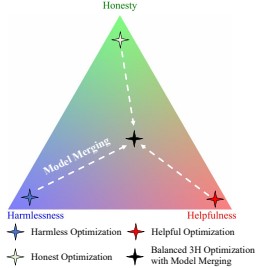

Figure 1: Illustration of trade-offs in optimizing LLMs across the 3H objectives.

---

*Corresponding Authors.

39th Conference on Neural Information Processing Systems (NeurIPS 2025).

while harmlessness alignment can lead to dishonest responses [12]. This trade-off is illustrated in Figure 1, highlighting the need for systematic approaches to harmonize 3H objectives.

Traditional methods for enhancing 3H properties often rely on *data mixing* strategies assisted by empirically heuristic rules [13], multi-dimensional scoring via reward models [14], or alignment conflict metrics [15], where diverse datasets are combined to fine-tune a single model. While effective, these approaches face practical limitations: (i) data curation requires substantial domain expertise and computational resources [11, 16], and (ii) conflicting optimization signals during fine-tuning may complicate prioritization of alignment objectives without compromising others [15, 17]. As a cost-effective alternative, model merging has gathered great attention for LLM alignment through integrating parameters from specialized aligned models, addressing key challenges such as catastrophic forgetting after fine-tuning [18] and achieving robust reward models [19, 20]. However, for 3H optimization, the effectiveness and limitations of existing merging methods remain underexplored, especially considering the preference noise [21] and layer significance [22, 23] for LLM multi-objective alignment. While preliminary investigations exist [24], these are narrowly focused on constrained scenarios (e.g., multilingual) or employ partial evaluations of 3H dimensions [25] without systematic comparisons. This raises the central question to be explored:

> *Can we benchmark the model merging and data mixing techniques in 3H optimization for LLM alignment and explore the overlooked optimization principles specific to model merging?*

To address this question, we first establish a benchmark for 3H optimization in LLM alignment and systematically compare model merging and data mixing techniques for 3H optimization. Based on this benchmark, we reveal previously overlooked collaborative and conflicting relationships among the 3H dimensions and discuss the advantages and limitations of data mixture (data-level) and model merging (parameter-level) methods in mitigating conflicts for balanced 3H optimization. Additionally, we address the challenges of preference noise accumulation and layer sparsity adaptation in LLM multi-objective merging, proposing a novel reweighting-enhanced task singular merging (RESM) method via outlier weighting and sparsity-aware rank selection to further enhance balanced LLM alignment. In summary, our key contributions are as follows:

• We create the first benchmark for 3H optimization in LLM alignment and systematically compare model merging and data mixing, including our investigations into 15 representative methods (12 training-free model merging methods and 3 representative data mixture methods), 10 preference datasets associated with 5 annotation dimensions, 2 classific LLMs families, and 2 training settings.
• We reveal a range of previously overlooked optimization principles and insights for 3H optimization in LLM alignment. These include: different collaborative and conflict relationships among 3H dimensions, the superiority of model merging over data mixture methods, and the factors affecting the effect of model merging considering redundant parameters updates during post-training.
• Beyond holistic evaluation of existing model merging methods, we propose a novel reweighting-enhanced task singular vector merging algorithm adapted to the preference noise accumulation and layer sparsity during merging through outlier weighting and sparsity-aware rank selection. Extensive experiments verify its effectiveness in achieving balanced LLM alignment.

## 2   Related Work

**Model Merging for LLM Alignment.** Model merging has emerged as a parameter-level cost-effective technique for LLM alignment [18], addressing challenges across four aspects: (a) *Stabilizing reference policies* focuses on the over-optimization problems during the RL training. Weight-space averaging of models with varying initializations constructs robust policy ensembles [26], while dynamic trust-region updates [27] and online gradient fusion [28] help preserve foundational capabilities. (b) *Cross-model capability transfer* resolves architectural mismatches during knowledge fusion [29] through probabilistic token alignment [30], vertical domain adaptation [31], and subspace projection [32]. But persistent toxic parameter propagation [33, 34] remains a critical barrier, inducing biased representation transfer during integration. (c) *Avoiding forgetting after finetuning* develops gradient-aware selective merging [35], heterogeneous layer-wise merging [36, 37], and subspace-based merging [38] to mitigate the alignment tax or realign the model after fine-tuning for downstream tasks. (d) *Balancing multi-optimized objectives* employs linear interpolation of reward-tuned models

[39, 17, 19, 20] and Mixture of Experts (MoE) based expert routing [25] to approximate Pareto frontiers but lacks theoretical guarantees for subspace conflict analysis. Additionally, while location-based merging [40] identifies alignment-specific weights, its efficacy is heavily data-dependent. Notably, [24] provides preliminary insights into safety-utility trade-offs in cross-lingual scenarios, but none of these studies explore model merging's potential and limitations for 3H optimization.

**Data Mixture for LLM Alignment.** Compared with the pretraining data mixture in terms of different domains, LLM alignment aims to achieve a good trade-off between Helpfulness, Honesty, and Harmlessness (3H) regarding human preference [41–43, 7]. (a) *Empirical Methods* [44, 45] have explored the heuristic mixture strategies between helpful and safety-related data to mitigate the safety-utility. (b) *Reward Model-based Methods* train traditional Bradley-Terry models [46, 47] and multi-objective reward models, which are designed to score the data for capturing the complicated human preferences [48–51]. ArmoRM [14] is a representative development aiming to promote LLMs aligned with human-interpretable multi-objective demands like honesty and helpfulness. (c) *New Metric Methods* are initially designed to select preference data only from the quality and diversity dimensions [52, 53], Hummer [15] recently quantifies the conflict among preference datasets to balance diverse alignment objectives effectively. Different from these works that resolve the conflict from the data mixture perspective, we explore the parameter-level model merging solutions and select representative data mixture methods as comparisons to discuss their effectiveness.

## 3 Revisting Model Merging for Multi-Object Alignment Optimization

### 3.1 Preliminaries

The intersection of model merging and alignment optimization presents unique challenges and opportunities that warrant dedicated investigation [19, 20]. Given multiple models parameterized by $\theta^1, \theta^2, \cdots, \theta^n$, where each optimizes base model $\theta^0$ towards a different alignment objective, the alignment task vector set can be achieved by $\Delta = (\Delta^1, \cdots, \Delta^n) = (\theta^1 - \theta^0, \cdots, \theta^n - \theta^0)$. Existing merging methods related to LLM multi-objective alignment [18] can be concluded as follows.

Linear interpolation methods, such as Rewarded Soups [17] and Weight Average based methods (WARM [19] and WARP [20]), have demonstrated that simple weighted averaging of model parameters can be effective in learning the Pareto frontier of multiple objectives or achieving robust reward models and reward policies. The merged model can be achieved through: $\theta_{\text{merged}} = \sum_{i=1}^{n} w_i \theta^i$, where the $w = (w_1, w_2, \cdots, w_n)$ is defined as the interpolation weight related to adjustable preference.

Task-Vector (TV) based methods [54] integrate different parameter update directions (the alignment task vector $\Delta^i = \theta^i - \theta^0$) rather than full model parameters like linear interpolation. Advanced merging approaches like TIES [55], DARE [56], Breadcrumbs [57], and DELLA [58] explore many nuanced ways to identify and preserve crucial subspaces that capture different objectives and resolve the objective conflicts. In general, these methods can be expressed as $\theta_{\text{Merged}} = \theta^0 + \sum_{i=1}^{n} w_i m_i \odot (\theta^i - \theta^0)$, where $m_i \in \{0, 1\}^{|\theta|}$ is a binary mask and $\odot$ is the element-wise multiplication. Moreover, Model stock [59] identifies that model performance correlates strongly with proximity to the center of the weight space and proposes to approximate this optimal center point geometrically.

Task Singular Vector (TSV) based methods [60–62] point out that the *element-wise* mask often breaks the inherent row–column correlations in the weight matrix, potentially destroying a low-dimensional structure of the fine-tuned parameters critical for individual tasks. As an alternative, they exploit the low-rank structure of task vectors through *layer-wise* parameter conflict analysis. By performing Singular Value Decomposition (SVD) with low-rank approximation on top-k singular components of layer-wise task vectors, we can achieve compressed or truncated task matrix through $\text{SVD}_k(\theta_l^i - \theta_l^0) = \boldsymbol{U}_l^{(i)}[:, : k_{\text{fixed}}]\boldsymbol{S}_l^{(i)}\boldsymbol{V}_l^{(i)\top}[: k_{\text{fixed}}, :]$, where $U, S$, and $V$ are the left singular vectors, singular values, and right singular vectors, the $k_{\text{fixed}}$ represents the top-k selection of singular values.

$$\min_{\boldsymbol{U}_{l\perp}} \|\boldsymbol{U}_{l\perp} - \boldsymbol{U}_l\|_F \quad \text{s.t.} \quad \boldsymbol{U}_{l\perp}^{\top}\boldsymbol{U}_{l\perp} = \boldsymbol{I}, \tag{1}$$

$$\min_{\boldsymbol{V}_{l\perp}} \|\boldsymbol{V}_{l\perp} - \boldsymbol{V}_l\|_F \quad \text{s.t.} \quad \boldsymbol{V}_{l\perp}^{\top}\boldsymbol{V}_{l\perp} = \boldsymbol{I}, \tag{2}$$

$$\theta_{\text{Merged Layer}} = \theta_l^0 + \sum_{i=1}^{n} \boldsymbol{U}_{l\perp}^{(i)}[:, : k_{\text{fixed}}]\boldsymbol{S}_l^{(i)}\boldsymbol{V}_{l\perp}^{(i)\top}[: k_{\text{fixed}}, :]. \tag{3}$$

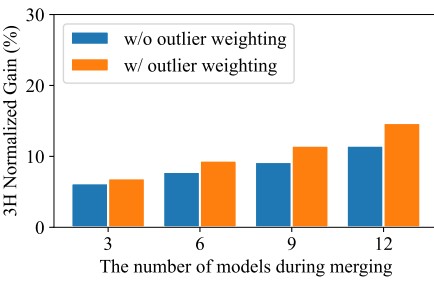 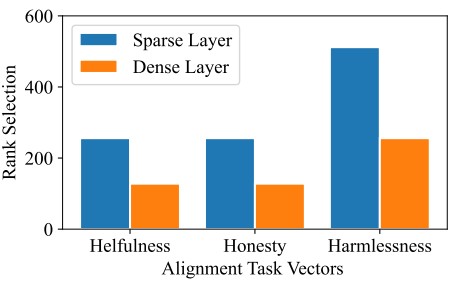

(a) Preference Noise Accumulation          (b) Effective Rank Analysis

Figure 2: (a) illustrates the impact of outlier weighting in model merging when incorporating multi-aligned models. The performance discrepancy before and after processing quantifies the degree of preference noise accumulation; (b) depicts the effective rank required to capture 95% of the total energy in singular values for task vectors derived from 3H training. Larger ranks indicate the necessity of retaining a higher proportion of singular values to preserve task-relevant information.

Assisted by decorrelat-based whitening transformation as Eq.(1) and Eq.(2), we can obtain orthogonal singular vectors $U_{l\perp}$ and $V_{l\perp}$ to mitigate the interference among task matrices. Finally, the layer-wise weight for the merged model can be defined as Eq.(3), where the $w_l$ is the scaling factor.

## 3.2 Motivation

Despite the successes of TSV-based methods outlined above, several overlooked challenges undermine the efficacy of model merging for 3H optimization:

**(i) Negative Outliers from Preference Noise Accumulation**: The core design of previous merging methods is to separate task-specific and task-shared parameters through element-wise mask or layer-wise SVD decomposition [18]. But all of these works ignore the preference noise accumulation [21], which is a special problem for LLM alignment, especially when more alignment objectives or models are considered during merging. This introduces additional outlier weight updates that make it difficult to capture true task-specific optimization direction, so as to weaken the effect of conflict resolution.

As shown in the Figure 2a, we respectively train models for each dimension of 3H optimization. The merging results with 3/6/9/12 models represent that for 3H optimization, we respectively select 1/2/3/4 single-dimension models with different hyperparameters. We apply the $3\sigma$ principle to mask the outlier weight for singular value parameter updates (the details can be shown in Section 4) while adopting the representative task-singular vector merging algorithm, TSVM [60]. From the results, we can find that masking the outlier weight within the singular value $S$ can strengthen the merging effect, and the accumulation preference noise has the same trend as the increased model numbers.

**(ii) Unreasonable Fixed Rank Selection for Each Layer**: Conventional model merging methods employ a uniform rank selection threshold $k$ across all layers, failing to account for layer-specific sparsity patterns and parameter importance heterogeneity in LLMs. As evidenced by layer-wise analyses [63], sparse attention activation and dense feed-forward layers exhibit fundamentally different sparsity characteristics. Moreover, recent mechanistic studies [23, 64, 22] reveal that alignment capabilities predominantly emerge from localized parameter updates in specific subnetwork regions rather than global changes. Both of them necessitate distinct rank selection strategies during merging.

As shown in the Figure 2b, we display the difference in the rank selection between dense and sparse layers while keeping 95 percent of information/energy within the singular value [65]. This divergence of rank among layers highlights the challenge of using a fixed top-k selection to compress the rank for model merging as Eq.(3), because we may either discard important components or keep unnecessary components that cause interference between different alignment objectives.

## 4    Reweighting-Enhanced Task Singular Vector Merging for 3H Optimization

**Overall:** Based on the above analysis, we propose a novel **R**eweighting **E**nhanced task **S**ingular vector **M**erging algorithm (RESM), with theoretical foundations in outlier detection and layer-wise

sparsity analysis for rank selection, aiming to improve model merging for 3H optimization. Thus, we can transform Eq.(3) to Eq.(4) by integrating with reweighting-based optimization. The full framework of RESM can be shown in Algorithm 1.

$$\boldsymbol{\theta}_l^{\text{RESM}} = \boldsymbol{\theta}_l^0 + \sum_{i=1}^n \underbrace{\boldsymbol{U}_{l\perp}^{(i)}[:,:k_l]}_{\substack{\text{Adaptive} \\ \text{Rank Selection}}} \underbrace{\left( \alpha_l^{(i)} \boldsymbol{S}_l^{(i)} \right)}_{\substack{\text{Outlier-Aware} \\ \text{Weighting}}} \underbrace{\boldsymbol{V}_{l\perp}^{(i)\top}[:k_l,:]}_{\substack{\text{Adaptive} \\ \text{Rank Selection}}}, \tag{4}$$

**To address (i)**, compared with directly utilizing full singular values, we employ layer-wise outlier detection to aggregate the Outlier-Aware Weight for reweighting the singular values. This means we should check which parts of the singular values can truly represent the optimization direction towards the alignment process. Considering the heavy-tailed distribution of LLM parameter updates, where few parameters undergo significant changes while most exhibit minor adjustments, we leverage the $3\sigma$ principle, which aligns with this characteristic. Thus, we adopt statistical significance filtering to weaken the noise effect and normalize competitive weight to identify true optimization adjustments. The $\Delta_{l,r,c}^{(i)} \in \mathbb{R}$ denotes the weight deviation at row $r$, column $c$ of layer $l$ for model $i$ relative to initial model, $\mu_r^{(i)}$ and $\sigma_r^{(i)}$ represent the mean (Eq.(5)) and standard deviation of deviations (Eq.(6)) in row $r$, quantifying central tendency and dispersion, $\alpha_l^{(i)} \in [0,1]$ (Eq.(7)) computes layer-wise aggregation weights via $L_1$-normalized sparse outlier magnitudes, and $\text{THRESHOLD}(\boldsymbol{M}, \tau)$ applies hard-thresholding to suppress elements in matrix $\boldsymbol{M}$ with absolute values below $\tau$.

$$\mu_{l,r}^{(i)} = \mathbb{E}_c[|\Delta_{l,r,c}|], \tag{5}$$

$$\sigma_{l,r}^{(i)} = \sqrt{\mathbb{E}_c[|\Delta_{l,r,c}|^2] - (\mu_{l,r}^{(i)})^2}, \tag{6}$$

$$\alpha_l^{(i)} = \frac{\sum_{r=1}^{d_l} \|\text{Threshold}(\boldsymbol{\Delta}_{l,r,:}^{(i)}, \mu_{l,r}^{(i)} + 3\sigma_{l,r}^{(i)})\|_1}{\sum_{j=1}^n \sum_{r=1}^{d_l} \|\text{Threshold}(\boldsymbol{\Delta}_{l,r,:}^{(j)}, \mu_{l,r}^{(j)} + 3\sigma_{l,r}^{(j)})\|_1} \tag{7}$$

Our outlier-aware reweighting mechanism operates through two complementary mechanisms: *Noise Suppression*: By thresholding parameter deviations via the $3\sigma$ rule, we filter out low-magnitude fluctuations that predominantly encode noise, forcing the singular vectors $\boldsymbol{u}_r^{(i)}$ to align with statistically significant task features; *Task Equilibrium*: The layer-wise aggregation weights $\alpha_l^{(i)}$ are globally normalized across all models, ensuring balanced contributions from diverse tasks and preventing dominance by high-magnitude updates that may obscure subtle yet critical features. More details for outlier-aware weighting for singular value can be shown in Appendix A.1.

**To address (ii)**, instead of fixed top-k strategy, we propose to adaptively decide the level of rank selection based on the layer sparsity. We can first compute the sparsity consensus for all models as Eq.(8) and then achieve the dynamic rank as Eq.(9), where $\gamma_0 > 0$ and $\gamma > 0$ control the base rank and sparsity-related rank reduction respectively, the $k_l$ is defined as the dynamic rank for layer $l$ determined by sparsity $\Omega_l$. We set $\gamma_0 = 0.2$, $\gamma = 0.6$ and $\epsilon = 0.1$ by default. We can observe that for layers with high sparsity ($\Omega_l \to 1$), the optimal rank selection $\Omega_l \to d_l(\gamma_0 + \gamma\Omega_l)$ retains most singular, whereas for dense layers $\Omega_l \to 0$, the rank $k_l$ decreases significantly, inducing stronger dimensionality reduction through truncation.

$$\Omega_l = \frac{1}{nd_l^2} \sum_{i=1}^n \sum_{r,c=1}^{d_l} \mathbb{I}\left(|\Delta_{l,r,c}^{(i)}| < \epsilon\right) \tag{8}$$

$$k_l = \lfloor d_l(\gamma_0 + \gamma\Omega_l) \rfloor \tag{9}$$

Our sparsity-adaptive rank selection mechanism operates through two complementary principles: *Information Preservation*: For dense layers, where parameter updates demonstrate relatively uniform distributions with predominantly small adjustments, employing lower-rank approximations proves effective for noise suppression while preserving principal components. However, this necessitates

**Algorithm 1:** Reweighting-Enhanced Task Singular Vector Merging

---

**Input** : Initial model $\boldsymbol{\theta}^0$ and Further Aligned models $\{\boldsymbol{\theta}^i\}_{i=1}^n$ with same layers $L$, Sparsity factor
$\qquad \gamma \in [0,1], \epsilon > 0$
**Output :** Merged model $\boldsymbol{\theta}^*$
**for** *layer* $l \leftarrow 1$ **to** $L$ **do**
$\quad$ // Step1:Alignment Task Vector Extraction
$\quad \boldsymbol{\Delta}_l^{(1:n)} \leftarrow [\boldsymbol{\theta}_l^i - \boldsymbol{\theta}_l^0]_{i=1}^n$
$\quad$ // Step2:Outlier-Aware Weighting
$\quad$ **for** *model* $i \leftarrow 1$ **to** $n$ **do**
$\quad\quad$ Compute row-wise statistics: $\boldsymbol{\mu}_l^{(i)}, \boldsymbol{\sigma}_l^{(i)} \leftarrow \text{ROWOUTLIERSCORE}(|\boldsymbol{\Delta}_l^{(i)}|)$
$\quad\quad$ Calculate sparse aggregation weights:
$\quad\quad \alpha_l^{(i)} \leftarrow \dfrac{\sum_{r=1}^{d_l} \|\text{THRESHOLD}(\boldsymbol{\Delta}_{l,r,:}^{(i)}, \mu_{l,r}^{(i)} + 3\sigma_{l,r}^{(i)})\|_1}{\sum_{j=1}^{n} \sum_{r=1}^{d_l} \|\text{THRESHOLD}(\boldsymbol{\Delta}_{l,r,:}^{(j)}, \mu_{l,r}^{(j)} + 3\sigma_{l,r}^{(j)})\|_1}$
$\quad$ // Step 3: Sparsity-Adaptive Rank Selection
$\quad$ Compute layer sparsity consensus: $\Omega_l \leftarrow \frac{1}{nd_l^2} \sum_{i=1}^n \sum_{r,c=1}^{d_l} \mathbb{I}(|\Delta_{l,r,c}^{(i)}| < \epsilon)$
$\quad$ Determine dynamic rank: $k_l \leftarrow \lfloor d_l(\gamma_0 + \gamma\Omega_l) \rfloor$
$\quad$ // Step 4:Reweighting Optimization during Merging
$\quad$ **for** *model* $i \leftarrow 1$ **to** $n$ **do**
$\quad\quad$ Decompose: $[\boldsymbol{U}_l^{(i)}, \boldsymbol{S}_l^{(i)}, \boldsymbol{V}_l^{(i)}] \leftarrow \text{SVD}(\boldsymbol{\Delta}_l^{(i)})$
$\quad\quad$ Compute orthogonal projections $\boldsymbol{U}_{l\perp}^{(i)}$ and $\boldsymbol{V}_{l\perp}^{(i)}$ via Eq.(1) via Eq.(2)
$\quad\quad$ Reweight for Outlier Weight: $\boldsymbol{S}_l^{(i)} \leftarrow \alpha_l^{(i)} \cdot \boldsymbol{S}_l^{(i)}$
$\quad\quad$ Reweight for Rank Selection: $\boldsymbol{U}_{l\perp}^{(i)} \leftarrow \boldsymbol{U}_{l\perp}^{(i)}[:, :k_l], \boldsymbol{V}_{l\perp}^{(i)} \leftarrow \boldsymbol{V}_{l\perp}^{(i)}[:k_l, :],$
$\quad\quad \boldsymbol{S}_l^{(i)} \leftarrow \boldsymbol{S}_l^{(i)}[:k_l, :k_l]$
$\quad$ Merge Components: $\boldsymbol{M}_l \leftarrow \sum_{i=1}^n \boldsymbol{U}_{l\perp}^{(i)} \boldsymbol{S}_l^{(i)} \boldsymbol{V}_{l\perp}^{(i)\top}$
$\quad$ Update the Layer for the Merged Model: $\boldsymbol{\theta}_l^* \leftarrow \boldsymbol{\theta}_l^0 + \boldsymbol{M}_l$

---

careful determination of the optimal rank selection threshold to balance between information retention and noise elimination. Conversely, sparse layers display concentrated parameter updates along a few dominant directions, potentially containing critical outlier components. Here, maintaining a higher rank becomes essential to ensure the preservation of these salient directional features, thereby preventing substantial information loss through excessive rank truncation. *Conflict Mitigation*: By preserving dominant singular directions in sparse layers and enforcing orthogonality through Eq.(1), we reduce overlaps between task-specific parameters, decoupling interference-prone optimization trajectories. More details about rank selection can be shown in Appendix A.2.

## 5 Experiments

### 5.1 Experimental Setup

**Datasets:** As shown in Table 1, we select commonly used preference data for model training. These datasets can be categorized into five groups from the annotation perspective.

**Backbones:** Following SimPO [72], we adopt two instruction-tuned models: Llama-3-8B-Instruct [73] and Mistral-7B-Instruct-v0.2 [74]. These serve as the SFT model, and we then perform DPO training on the full network using the preference data.

**Baselines:(i) Individual Training**: We respectively train one model for each annotation perspective as stated by Table 1. These models are saved for model merging. **(ii) Mixture Training**: We adopt the full datasets shown in Table 1 and then adjust the data mixture proportion before

Table 1: Dataset statistics for our DPO training.

| Annotation Perspective | Dataset | Judge |
|---|---|---|
| Helpfulness | HelpSteer [49] | GPT4-Turbo |
| | Py-Dpo [66] | GPT4-Turbo |
| | Distilabel-Orca [67] | GPT4-Turbo |
| | Distilabel-Capybara [68] | GPT4-Turbo |
| Harmlessness | UltraSafety [5] | GPT4-Turbo |
| Honesty | Truthy-Dpo-v0.1 [69] | Human |
| | GRATH [70] | Llama2 |
| Helpfulness&Honesty | UltraFeedback [52] | GPT4-Turbo |
| Helpfulness&Harmlessness | PKU-Safe-RLHF [11] | GPT4-Turbo |
| | Nectar [71] | GPT4-Turbo |

training based on the *Empirical* methods [44, 45], the multi-dimension score of the Reward model *ArmoRM-Llama3-8B* [14] and the alignment conflict metric from *Hummer* [15]. The implementation details can be shown in Appendix C;**(iii) Model Merging:** Considering that the constraints of data availability and test data leak will limit the generalization of merging methods for LLMs, we mainly adopt *training-free multi-task or multi-object alignment* merging strategies from MergeKit [75], which includes Weight Average [76], Task Arithmetic [54], Ties-Merging [55], DARE[56], DELLA [58], Model Stock [59] and Model Breadcrumbs [57]. Moreover, from the perspective of Pareto-optimal front [39, 17, 19, 20] and singular vector decomposition [60, 61], we select Rewarded Soup [17], TSVM [60] as two additional training-free merging methods for 3H optimization. We provide discussions about training-based and MOE-based merging methods [25] in Appendix B.

**Settings:** We construct two different settings to verify the effectiveness of model merging for 3H optimization: **(i) Static Optimization for DPO Training at once** as Table 3 and Table 4, where we aim to achieve an aligned model that simultaneously meets the 3H demands using various annotated preference data at once. **(ii) Continual Optimization for Sequential DPO Training** as Table 10 and Table 11, which refers to the continual and dynamic circumstances with newly curated preference data and more customized demands compared to previously trained models. In this case, we need to simultaneously focus on the effectiveness and efficiency of constructing an aligned model.

**Evaluation: (i) For Helpfulness:** we select Math, GSM8K, ARC-C, ARC-E, MMLU, MBPP-Plus, HumanEval-Plus [77], and MT-Bench [78] to asses the helpfulness of LLMs; **(ii) For Honesty:** we utilize the HaluEval-Wild [79] for evaluating honesty or hallucinations; **(iii) For Harmlessness:**, we conduct safety-related (SaladBench [80]) and refusal-related (OR-Bench [81]) evaluations to measure the harmlessness of models. Higher values are preferred for all reported results to ensure the reasonableness and fairness of evaluation. The *normalized metric* is calculated based on the relative gain of each dimension to avoid the imbalanced evaluation datasets for the 3H perspective. More details can be shown in the Appendix C.2.

Table 2: Necessary specifications for the strategy and scaling of each method.

| Method | Strategy | Scaling |
|---|---|---|
| **Data Mixture-Based Methods** | | |
| Heuristic [13] | Empirically heuristic-adjusted ratio | Data Mixture Ratio |
| ArmoRM [14] | Reward Model | Multi-object Data Selection |
| Hummer [15] | Alignment Conflict Metric | Multi-object Data Selection |
| **Merging-Based Methods** | | |
| Weight Average [76] | Linear Int. Consensus | Parameter Weight Coeff. |
| Rewarded Soup [17] | Linear Int. Consensus | Parameter Weight Coeff. |
| Task Arithmetic [54] | Linear Int. Consensus | Parameter Scaling Factor |
| Ties [55] | Top-k Sparsification | Parameter Scaling Factor |
| DARE [56] | Random Sparsification | Parameter Scaling Factor |
| DELLA [58] | Random Sparsification | Parameter Scaling Factor |
| Breadcrumbs [57] | Top/Bottom-k Sparsification | Parameter Scaling Factor |
| Model Stock [59] | Geometric Sparsification | Parameter Adaptive Ratio |
| TSVM [60] | Singular Value Decomposition | Parameter Scaling Factor |

## 5.2 Experimental Results

**There exist different collaborative and conflict relationships in terms of 3H objectives for LLM alignment.** As shown in Figure 3, we display the trade-off through individual training comparison. Denote the results of Instruct LLMs as the grey line in the Figure, we can compare the results of Honesty and Helpfulness after performing individual Helpful, Honest, and Harmless Training to distinguish the relationship between each optimization dimension. From the results, we can observe that there exist different collaborative and conflict relationships between helpfulness, honesty, and harmlessness while performing DPO Training, exhibiting that Helpful Training benefits 3H performance simultaneously, but Honest and Harmless Training weaken each other.

**Model merging can serve as a good alternative for data mixture methods in mitigating the 3H conflict.** As shown in Table 3 and Table 4, we compare the effectiveness of data mixture and model merging methods for 3H optimization of Llama3 [73] and Mistral [74]. *For data mixture methods*, they mitigate the 3H conflict by collecting full training preference data and then filtering conflict samples. Compared with heuristic strategies and reward model based strategies, which rely on historical training data and human effort, Hummer is specially designed for evaluating conflict for full training preference data, which can achieve better 3H results with a norm gain from 7.68% to 10.16% on the Llama3. *For model merging methods*, compared with full training strategies, they advocates for phased optimization to negotiate competing alignment objectives through *dimension-specific individual training*, where we first conduct individual training to obtain models for five different annotation dimensions respectively, and then adopt *conflict-aware parameter merging strategies*, such as random sparsification, Top-$k$ filtering, and singular value decomposition, to merge these models into an ideal one that can achieve close or superior results than full training methods. Take the experiment on Llama3 for example, compared with the best data mixture methods, Hummer(10.16%), model

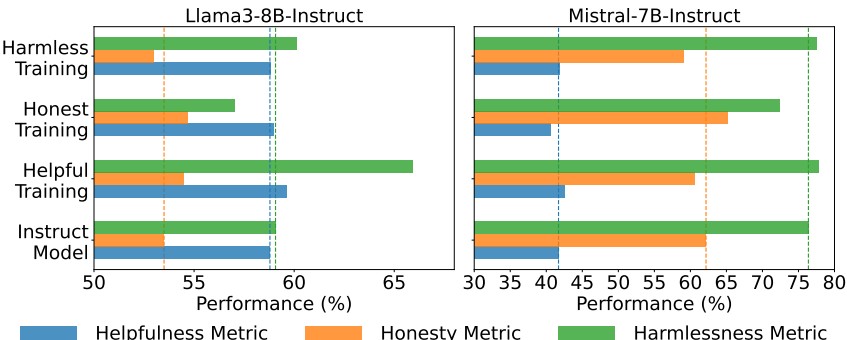

Figure 3: Illustration of the 3H Trade-Off, where Helpful Training benefits 3H performance simultaneously, but Honest and Harmless Training weaken each other. The dashed line (Instruct Model) serves as the reference, where rectangles on the left represent the weakening effect, and vice versa.

Table 3: 3H Results Under Static Optimization Setting where we perform DPO training using various datasets at once. The normalized gain metric is the average value of the relative gain for each dimension compared with the results of Llama3-8B-Instruct.

| Methods | Helpfulness | | | | | | | | Honesty | Harmlessness | | Helpful_Avg | Honest_Avg | Harmless_Avg | Norm_Gain |
|---|---|---|---|---|---|---|---|---|---|---|---|---|---|---|---|
| | Math | GSM8K | ARC-E | ARC-C | MMLU | MBPP_Plus | HumanEval_Plus | MT-Bench | HaluEval_Wild | Salad_Bench | OR-Bench | | | | |
| **llama3-8B-Instruct** | 28.08 | 78.09 | 93.65 | 82.03 | 68.20 | 58.99 | 53.05 | 8.25 | 53.50 | 91.16 | 26.97 | 58.79 | 53.50 | 59.07 | — |
| Helpfulness | 29.08 | 79.30 | 93.65 | 81.69 | 68.58 | 57.94 | 58.54 | 8.33 | 55.00 | 89.83 | 42.06 | 59.64↑1.45% | 55.00↑2.80% | 65.95↑11.65% | +5.30% |
| Honesty | 28.52 | 78.77 | 93.65 | 81.36 | 68.34 | 58.47 | 54.27 | 8.45 | 54.67 | 92.18 | 21.95 | 58.98↑0.32% | 54.67↑2.19% | 57.07↓3.39% | -0.29% |
| Harmlessness | 28.88 | 77.33 | 93.65 | 82.03 | 68.32 | 59.79 | 52.44 | 8.15 | 53.33 | 92.36 | 27.92 | 58.82↑0.05% | 53.33↓0.32% | 60.14↑1.81% | +0.51% |
| Helpfulness&Honesty | 29.60 | 77.63 | 93.47 | 82.71 | 68.33 | 59.79 | 59.15 | 8.18 | 56.00 | 90.86 | 39.80 | 59.86↑1.82% | 56.00↑4.67% | 65.33↑10.60% | +5.70% |
| Helpfulness&Harmlessness | 30.02 | 77.26 | 93.47 | 82.37 | 68.31 | 58.99 | 56.11 | 8.16 | 54.50 | 90.27 | 58.86 | 59.34↑0.94% | 54.50↑1.87% | 74.57↑26.24% | +9.68% |
| 3H Mixture Full Training (Heurisistic) | 28.21 | 78.85 | 93.65 | 81.69 | 68.38 | 60.85 | 57.32 | 8.48 | 54.67 | 92.06 | 35.36 | 59.68↑1.51% | 54.67↑2.19% | 63.71↑7.85% | +3.85% |
| 3H Mixture Full Training (ArmoRM) | 28.81 | 78.97 | 93.65 | 82.39 | 68.42 | 60.55 | 58.22 | 8.52 | 55.50 | 92.11 | 42.12 | 60.24↑2.47% | 55.50↑3.74% | 69.02↑16.84% | +7.68% |
| 3H Mixture Training (Hummer) | 29.41 | 78.95 | 93.65 | 82.69 | 68.59 | 60.41 | 58.15 | 8.58 | 55.60 | 92.10 | 50.11 | 60.35↑2.65% | 55.60↑3.93% | 73.21↑23.90% | +10.16% |
| Weight Average | 29.80 | 78.01 | 93.47 | 82.71 | 68.43 | 59.26 | 57.32 | 8.02 | 57.78 | 91.72 | 41.48 | 59.63↑1.43% | 57.78↑8.00% | 66.60↑12.75% | +7.39% |
| Rewarded Soup | 29.64 | 77.94 | 93.47 | 82.71 | 68.54 | 60.85 | 57.93 | 8.32 | 57.55 | 90.86 | 50.08 | 59.93↑1.94% | 57.55↑7.57% | 70.47↑19.30% | +9.60% |
| Model Stock | 28.72 | 78.24 | 93.47 | 82.71 | 68.41 | 59.79 | 56.10 | 8.03 | 53.00 | 91.62 | 32.28 | 59.43↑1.09% | 53.00↓0.93% | 61.95↑4.88% | +1.68% |
| Task Arithmetic | 29.02 | 79.05 | 93.30 | 83.39 | 68.35 | 57.14 | 51.83 | 8.37 | 57.33 | 91.39 | 28.29 | 58.81↑0.03% | 57.33↑7.16% | 59.84↑1.30% | +2.83% |
| Ties | 29.30 | 78.17 | 93.30 | 83.05 | 68.52 | 56.61 | 53.05 | 8.20 | 54.33 | 89.13 | 29.07 | 58.78↓0.02% | 54.33↑1.55% | 59.10↑0.05% | +0.49% |
| DARE | 29.42 | 78.39 | 93.47 | 82.71 | 68.41 | 59.26 | 56.71 | 8.28 | 57.00 | 91.85 | 38.80 | 59.58↑1.34% | 57.00↑6.54% | 65.33↑10.60% | +6.16% |
| DARE Ties | 29.64 | 78.01 | 93.47 | 82.71 | 68.43 | 59.26 | 57.32 | 8.07 | 56.00 | 92.16 | 36.88 | 59.61↑1.40% | 56.00↑4.67% | 64.52↑9.23% | +5.10% |
| DELLA | 29.08 | 78.92 | 93.30 | 83.73 | 68.41 | 54.76 | 52.44 | 8.43 | 54.67 | 91.37 | 63.31 | 58.63↑0.27% | 54.67↑2.19% | 77.34↑30.93% | +10.95% |
| DELLA Ties | 29.20 | 75.97 | 93.30 | 83.39 | 68.46 | 56.08 | 49.39 | 8.16 | 54.00 | 87.95 | 70.02 | 57.99↑1.36% | 54.00↓0.93% | 78.99↑33.72% | +11.10% |
| Breadcrumbs | 29.46 | 77.79 | 93.30 | 83.39 | 68.53 | 60.85 | 54.88 | 8.24 | 59.33 | 91.60 | 44.85 | 59.56↑1.31% | 59.33↑10.89% | 68.23↑15.51% | +9.24% |
| Breadcrumbs Ties | 29.72 | 78.54 | 93.30 | 83.05 | 68.46 | 60.05 | 53.66 | 8.14 | 55.50 | 90.00 | 58.52 | 59.37↑0.99% | 55.50↑3.74% | 74.26↑25.72% | +10.15% |
| TSVM | 29.92 | 77.63 | 93.12 | 82.17 | 68.51 | 59.26 | 55.49 | 8.29 | 56.20 | 89.43 | 67.76 | 59.30↑0.87% | 56.20↑5.05% | 78.60↑33.08% | +13.00% |
| **RESM (ours)** | 29.89 | 78.77 | 93.65 | 82.27 | 68.41 | 59.46 | 56.55 | 8.29 | 58.20 | 89.92 | 69.27 | 59.62↑1.41% | 58.20↑8.79% | 79.60↑34.72% | **+14.97%** |

merging methods including DELLA Ties(11.10%), Breadcrumbs Ties(10.15%), TSVM(13.00%) ,and our RESM(14.97%) can consistently achieve comparable or superior results for balanced 3H optimization. These results collectively confirm that model merging's phased optimization paradigm effectively negotiates competing alignment objectives, which provides new insights for addressing the trilemma of 3H optimization for LLM alignment.

**The effect of model merging for 3H optimization is closely related to their conflict-resolution strategies. RESM consistently achieves better results due to its reweighting designs.** As shown in Table 2, we can divide existing parameter-level strategies into three categories: linear consensus, sparsification, and singular value decomposition. Linear interpolation of full model parameters or task vectors neglects the parameter conflict, limiting their performance for 3H optimization in LLM alignment. The sparsification-based method holds the assumption that pruning redundant (DARE and DELLA) or outlier parameters (Breadcrumbs) that do not represent the direction of updates for task vectors can improve the effect of model merging, but the level of sparsity is difficult to control for LLM through even with different sparsification methods. From the results of Table 3 and Table 4, we can observe that there is no fixed and stable trend for the results of sparsification-based methods due to random sparsification. For example, DELLA-Ties and DARE-Ties exhibit opposite phenomena in Llama3 and Mistral. More details about the sparsity that influences the effect of merging can be shown in Appendix C.4. In contrast, both TSVM and RESM can achieve stable performance gain through decomposed task singular vectors without heavily depending on sparsification. Moreover, our RESM enhances the merging effect of TSVM with a normalized gain from 13.00% to 14.97% due to its reweighting optimization adapted to preference noise accumulation and fixed rank problems.

**RESM can achieve robust and efficient 3H optimization in continual LLM alignment than previous methods** As shown in Table 10 and Table 11 in the Appendix C.3, we sequentially perform DPO training using data with annotations about Helpfulness&Honesty (Stage1), Helpfulness&Harmlessness (Stage2) and Helpful (Stage3) to simulate continuous optimization in real-world

Table 4: 3H Results Under Static Optimization Setting where we perform DPO training using various datasets at once. The normalized gain metric is the average value of relative gain for each dimension compared with the results of Mistral-7B-Instruct-v0.2.

| Methods | Helpfulness | | | | | | | | Honesty | Harmlessness | | Helpful_Avg | Honest_Avg | Harmless_Avg | Norm_Gain |
|---|---|---|---|---|---|---|---|---|---|---|---|---|---|---|---|
| | Math | GSM8K | ARC-E | ARC-C | MMLU | MBPP_Plus | HumanEval_Plus | MT-Bench | HaluEval_Wild | Salad_Bench | OR-Bench | | | | |
| **Mistral-7B-Instruct-v0.2** | 9.54 | 46.17 | 82.36 | 72.88 | 59.97 | 26.46 | 28.66 | 7.55 | 62.17 | 78.07 | 74.68 | 41.70 | 62.17 | 76.38 | — |
| Helpfulness | 9.44 | 47.38 | 84.48 | 75.25 | 60.70 | 22.49 | 32.31 | 7.80 | 60.60 | 74.84 | 80.67 | 42.48↑1.87% | 60.60↓2.52% | 77.76↑1.81% | +0.39% |
| Honesty | 9.34 | 46.63 | 82.54 | 71.19 | 59.04 | 24.34 | 23.78 | 7.76 | 65.20 | 83.98 | 60.82 | 40.58↓2.69% | 65.20↑4.87% | 72.40↓5.21% | -1.01% |
| Harmlessness | 9.40 | 46.10 | 81.42 | 72.88 | 60.03 | 27.51 | 30.39 | 7.43 | 59.00 | 85.43 | 69.54 | 41.90↓0.48% | 59.00↓5.10% | 77.49↑1.45% | -1.06% |
| Helpfulness&Honesty | 8.76 | 43.14 | 82.01 | 74.92 | 59.78 | 25.93 | 27.33 | 7.59 | 61.33 | 78.74 | 77.23 | 41.18↓1.25% | 61.33↓1.35% | 77.99↑2.11% | -0.16% |
| Helpfulness&Harmlessness | 9.96 | 41.77 | 83.42 | 75.59 | 61.13 | 34.92 | 29.27 | 7.46 | 50.60 | 79.83 | 86.05 | 42.94↑2.97% | 50.60↓18.61% | 82.94↑8.59% | -2.35% |
| 3H Mixture Full Training (Heurisistic) | 9.56 | 41.70 | 83.06 | 74.24 | 60.23 | 22.75 | 34.15 | 7.69 | 62.00 | 81.13 | 72.97 | 41.67↓0.07% | 62.00↓0.27% | 77.05↓0.88% | +0.18% |
| 3H Mixture Full Training (ArmoRM) | 9.71 | 43.70 | 83.65 | 74.54 | 60.33 | 25.11 | 33.58 | 7.75 | 61.95 | 81.10 | 75.55 | 42.30↑1.44% | 61.95↓0.35% | 78.32↑2.54% | +1.21% |
| 3H Mixture Training (Hummer) | 9.79 | 44.50 | 83.72 | 74.89 | 60.53 | 25.85 | 33.15 | 7.56 | 62.05 | 81.85 | 75.28 | 42.50↑1.92% | 62.05↓0.19% | 78.57↑2.87% | +1.53% |
| Weight Average | 9.86 | 45.49 | 82.36 | 74.58 | 60.70 | 27.25 | 31.10 | 7.47 | 60.60 | 81.51 | 72.90 | 42.35↑1.56% | 60.60↓2.52% | 77.21↑1.09% | +0.04% |
| Rewarded Soup | 9.74 | 45.11 | 82.54 | 74.58 | 60.65 | 26.72 | 30.49 | 7.48 | 60.71 | 81.43 | 72.51 | 42.16↑1.10% | 60.71↓2.35% | 76.97↑0.77% | -0.16% |
| Model Stock | 9.80 | 46.93 | 82.36 | 74.24 | 60.34 | 25.13 | 30.49 | 7.19 | 61.81 | 80.11 | 73.78 | 42.06↑0.86% | 61.81↓0.58% | 76.95↑0.75% | +0.34% |
| Task Arithmetic | 9.76 | 44.12 | 84.13 | 73.90 | 60.86 | 27.25 | 33.54 | 7.44 | 61.01 | 83.91 | 71.04 | 42.63↑2.23% | 61.01↓1.87% | 77.48↑1.44% | +0.60% |
| Ties | 10.22 | 41.47 | 85.19 | 74.92 | 61.34 | 27.25 | 31.10 | 7.46 | 58.73 | 82.15 | 81.13 | 42.37↑1.61% | 58.73↓5.53% | 81.64↑6.89% | +0.99% |
| DARE | 10.10 | 43.82 | 84.48 | 73.90 | 60.78 | 27.25 | 32.93 | 7.47 | 61.05 | 83.80 | 71.04 | 42.59↑2.13% | 61.05↓1.80% | 77.42↑1.36% | +0.56% |
| DARE Ties | 10.10 | 42.46 | 85.00 | 74.58 | 61.05 | 26.98 | 31.71 | 7.61 | 59.28 | 82.28 | 81.91 | 42.49↑1.89% | 59.28↓4.65% | 82.10↑7.49% | +1.58% |
| DELLA | 10.26 | 42.76 | 84.83 | 73.56 | 60.86 | 26.19 | 32.93 | 7.47 | 61.00 | 84.36 | 77.20 | 42.35↑1.56% | 61.00↓1.88% | 77.78↑1.83% | +0.50% |
| DELLA Ties | 10.14 | 40.71 | 84.48 | 75.93 | 61.56 | 30.95 | 32.32 | 7.40 | 56.10 | 82.00 | 84.27 | 42.94↑2.97% | 56.10↓9.76% | 83.14↑8.85% | +0.69% |
| Breadcrumbs | 9.48 | 43.06 | 83.60 | 73.56 | 60.94 | 27.25 | 31.71 | 7.52 | 60.20 | 83.88 | 70.91 | 42.14↑1.06% | 60.20↓3.17% | 77.40↑1.34% | -0.26% |
| Breadcrumbs Ties | 10.20 | 41.85 | 85.01 | 76.61 | 61.27 | 26.72 | 30.49 | 7.48 | 60.04 | 81.83 | 80.49 | 42.45↑1.80% | 60.04↓3.43% | 81.16↑6.26% | +1.54% |
| TSVM | 10.40 | 44.88 | 84.29 | 75.24 | 60.87 | 28.50 | 32.32 | 7.65 | 61.10 | 83.25 | 78.51 | 43.02↑3.17% | 61.10↓1.72% | 80.88↑5.89% | +2.45% |
| **RESM (ours)** | 10.44 | 45.00 | 84.35 | 75.79 | 60.87 | 28.50 | 32.52 | 7.71 | 61.50 | 84.25 | 80.25 | 43.15↑3.48% | 61.50↓1.08% | 82.25↑7.69% | **+3.36%** |

scenarios. Comparative analysis of continual training stages reveals that catastrophic forgetting effects paradoxically enhance large language models' (LLMs) alignment with honesty, helpfulness, and harmlessness (3H) through interactive optimization. Our method strategically initializes model merging from the foundational Instruct checkpoint rather than intermediate checkpoints, thereby eliminating hyperparameter sensitivity in continual DPO training while mitigating overfitting to prior objectives that impede adaptation to new optimization targets. Experimental results demonstrate RESM's superior performance over final-stage models across 3H metrics, confirming its robustness for multi-objective alignment optimization.

**RESM can achieve better results than traditional sparse LLM works with outlier-based optimization.** While outlier-based optimization can also be used for pruning LLMs [84, 85], they are *two-stage* methods constrained to dealing with the parameters of only one LLM, while we focus on the outlier weights of different LLMs during *end-to-end* model merging. This means we should additionally consider the parameters conflict while merging different LLMs, rather than post-hoc process a well-merged LLM. To further distinguish our contribution, as shown in Table 5, we conduct outlier-related experiments on Llama3, including two representative outlier-based pruning methods, Wanda [82] and Owl [83]. We can observe that RESM can consistently achieve better results.

Table 5: Comparison with other outlier-based LLM works on Llama3.

| Method | Helpfulness | Honesty | Harmlessness | Norm_Gain |
|---|---|---|---|---|
| Llama3-8B-Instruct | 58.79 | 53.50 | 59.07 | – |
| TSVM | 59.30↑0.87% | 56.20↑5.05% | 78.60↑33.06% | +12.99% |
| TSVM+Wanda [82] | 59.35↑0.95% | 56.32↑5.27% | 78.75↑33.28% | +13.17% |
| TSVM+Owl[83] | 59.41↑1.05% | 56.40↑5.42% | 79.15↑33.93% | +13.47% |
| RESM (ours) | 59.62↑1.41% | 58.20↑8.79% | 79.60↑34.72% | **+14.97%** |

**Ablation Studies**. As shown in Table 6 and Table 7, we can observe that integrating both outlier weighting and sparsity-adaptive rank selection can collectively enhance the merging effect, which can verify the responsibility of our reweighting-based optimization. Notably, RESM outperforms data mixture baselines, achieving a gain of close to 1.5x to 2.1x improvement. These results validate model merging as a viable pathway for LLM alignment towards balancing multi-dimensional objectives

Table 6: Ablation studies for Reweighting-Induced Improvements on Llama3.

| Method | Helpfulness | Honesty | Harmlessness | Norm_Gain |
|---|---|---|---|---|
| Llama3-8B-Instruct | 58.79 | 53.50 | 59.07 | – |
| Hummer (best mixture) | 60.35↑2.65% | 55.60↑3.93% | 73.21↑23.94% | +10.17% |
| TSVM (best merging) | 59.30↑0.87% | 56.20↑5.05% | 78.60↑33.06% | +12.99% |
| RESM w/o Outlier Weighting | 59.52↑1.24% | 56.20↑5.05% | 79.45↑34.51% | +13.60% |
| RESM w/o Rank Selection | 59.45↑1.12% | 56.80↑6.17% | 79.05↑33.83% | +13.71% |
| RESM (ours) | 59.62↑1.41% | 58.20↑8.79% | 79.60↑34.72% | **+14.97%** |

Table 7: Ablation studies for Reweighting-Induced Improvements on Mistral.

| Method | Helpfulness | Honesty | Harmlessness | Norm_Gain |
|---|---|---|---|---|
| Mistral-7B-Instruct-v0.2 | 41.70 | 62.17 | 76.38 | – |
| Hummer (best mixture) | 42.50↑1.92% | 62.05↓0.19% | 78.57↑2.87% | +1.53% |
| TSVM (best merging) | 43.02↑3.17% | 61.10↓1.72% | 80.88↑5.89% | +2.45% |
| RESM w/o Outlier Weighting | 42.90↑2.88% | 61.80↓0.60% | 81.25↑6.38% | +2.89% |
| RESM w/o Rank Selection | 43.32↑3.89% | 61.20↓1.56% | 81.75↑7.03% | +3.12% |
| RESM (ours) | 43.15↑3.48% | 61.50↓1.08% | 82.25↑7.69% | **+3.36%** |

## 6 Conclusion

This paper establishes the first benchmark to systematically compare data mixture and model merging methods for balanced optimization across helpfulness, harmlessness, and honesty dimensions to enhance LLMs' alignment. Leveraging this benchmark, we uncover a series of overlooked optimization principles and insights. Specifically, we propose a novel Reweighting-Enhanced Task Singular Merging (RESM) method, which employs outlier weighting and sparsity-aware rank selection strategies to address preference noise accumulation and layer sparsity adaptation challenges during LLM merging for 3H objectives. Our theoretical analyses and experimental results provide a promising pathway for LLM alignment, advancing the development of ethically constrained language models.

## ACKNOWLEDGMENTS

This work was supported in part by the National Key Research and Development Program of China (2024YFE0203700), "Pioneer" and "Leading Goose" R&D Program of Zhejiang (2025C02037), and the Starry Night Science Fund of Zhejiang University Shanghai Institute for Advanced Study (SN-ZJU-SIAS-0010). Moreover, this work was also supported by Ant Group Research Intern Program.

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

# A   More Details for Method

## A.1   More Details for Outlier-Aware Weighting

**Interpretation of Dual Objectives for outlier weighting**   The mathematical framework achieves cross-model consensus and intra-model saliency through its hierarchical thresholding mechanism:

(i) **Cross-Model Consensus**: The denominator in Eq. (3) normalizes each model's contribution by the total sparse outlier magnitude across all $n$ models:

$$\sum_{j=1}^{n} \sum_{c=1}^{d_l} \|\text{THRESHOLD}(\mathbf{\Delta}_{l,c}^{(j)}, \mu_c^{(j)} + 3\sigma_c^{(j)})\|_1 \tag{10}$$

This forces models with greater sparse deviation magnitudes (potential task conflicts) to receive proportionally reduced aggregation weights $\alpha_l^{(i)}$, effectively suppressing outlier-dominated models in the merged output.

(ii) **Intra-Model Saliency**: The $3\sigma$ threshold in $\text{THRESHOLD}(\mathbf{\Delta}_{l,c}^{(i)}, \mu_c^{(i)} + 3\sigma_c^{(i)})$ implements statistical outlier detection within each model's parameter distribution. For Gaussian-distributed $\Delta_{l,c,k}^{(i)}$ (per Central Limit Theorem), this retains only the top 0.3% extreme deviations that likely correspond to:

- Task-specific knowledge carriers ($\Delta > \mu + 3\sigma$)
- Catastrophic interference sources ($\Delta < \mu - 3\sigma$)

The $L_1$ norm aggregation $\sum_{c=1}^{d_l} \|\cdot\|_1$ then amplifies layers containing concentrated outlier parameters.

**Synergistic Effect**: The normalization in (i) prevents any single model's outliers from dominating the merger, while the saliency detection in (ii) preserves critical task-specific features within each model. This dual mechanism reduces interference by selectively blending statistically significant parameters across models.

## A.2 More Details for Dynamic Rank Selection

Our method provides enhanced guarantees through statistical awareness and adaptive computation.

**(i) Conflict Probability Bound** Let $p_{\text{conflict}}^{(l)}$ denote the probability of directional conflicts in layer $l$. Our rank adaptation yields as follows. We can observe that , compared to TSVM's fixed $\frac{1}{\sqrt{d_l}}$, our bound adapts to layer sparsity.

$$\mathbb{E}[p_{\text{conflict}}^{(l)}] \leq \frac{1}{\sqrt{k_l}} \propto \frac{1}{\sqrt{\lfloor d_l(\gamma_0 + \gamma\Omega_l)\rfloor}} \tag{11}$$

**(ii)Theorem Proof for Conflict Probability Bound**: Let $(u, v) \in \mathbb{R}^{k_l}$ be random unit vectors representing Task A's and Task B's optimization direction after projection. The conflict probability is defined as follows, assisted by similarity, where $\epsilon$ is the conflict threshold (usually set as 0.3):

$$P_{\text{conflict}}^l = \mathbb{P}(\cos\theta > \epsilon) = \mathbb{P}(\langle u, v\rangle > \epsilon) \tag{12}$$

We can calculate the concentration onthe Hypersphere by Lévy's concentration lemma as follows:

$$\mathbb{P}(|\mathbf{u}^T\mathbf{v}| \geq t) \leq 2e^{-k_l t^2/2} \tag{13}$$

**For previous fixed rank**: We can bound the expected conflict probability:

$$\mathbb{E}[P_{\text{conflict}}^l] = \int_0^1 \mathbb{P}(|\mathbf{u}^T\mathbf{v}| \geq t)dt \tag{14}$$

$$= \int_0^\epsilon 2e^{-k_l t^2/2}dt + \int_\epsilon^1 2e^{-k_l t^2/2}dt \tag{15}$$

when $t \in (0, \epsilon)$, this term $\leq 2\epsilon$;

when $t \in (\epsilon, 1)$, this term $\leq 2(1-\epsilon)e^{-k_l\epsilon^2/2}$.

when we choose $\epsilon = \frac{1}{\sqrt{k_1}}$, it becomes:

$$\mathbb{E}[p_{\text{conflict}}] \leq \frac{2}{\sqrt{k_1}} + \frac{2}{e^{1/2}\sqrt{k_1}} \leq \frac{2.5}{\sqrt{k_1}}$$

**For our adaptation rank**, we can substitute the adaptive rank as follows:

$$k_1 = \lfloor d_l(\gamma_0 + \gamma\Omega_l)\rfloor$$

Thus, we can conclude:

$$\mathbb{E}[p_{\text{conflict}}] \leq \frac{2.5}{\sqrt{\lfloor d_l(\gamma_0 + \gamma\Omega_l)\rfloor}}$$

The key insight includes two parts:

- On the one hand, in $\mathbb{R}^{k_1}$, unit task vectors become increasingly orthogonal (evaluated by the dot product $|\mathbf{u}^\top\mathbf{v}|$) as $k_1 \to \infty$
- On the other hand, sparsity adaptation controls this effect

## A.3 Order of Orthogonalization and Rank Selection

A critical design in our RESM algorithm lies in the sequential relationship between orthogonalization (Eq. 1-2) and rank selection (Eq. 9). Through theoretical analysis and empirical validation, we establish that **orthogonalization should precede selection** to ensure optimal subspace alignment and information preservation. This ordering stems from three fundamental considerations below:

**(i) Global Orthogonality Constraints**: The orthogonal projection in Eq. 1 minimizes the Frobenius norm difference $\|U_{l\perp} - U_l\|_F$ under strict orthogonality constraints. Performing this projection

Table 8: Theoretical Comparison between our proposed RESM and TSVM.

| Property | TSVM | RESM |
|---|---|---|
| Layer adaptivity | $\times$ | $\checkmark$ |
| Sparsity awareness | $\times$ | $\checkmark$ |
| Conflict bound | $O(d^{-1/2})$ | $O(d^{-1/2}(\gamma_0 + \gamma\Omega_l)^{-1/2})$ |
| Weight concentration | Uniform | Heavy-tailed |
| Comp. complexity | $O(d^3)$ | $O(kd^2)$ |

*before* selection preserves the complete singular vector structure, enabling accurate modeling of cross-task interference patterns. Early selection would discard essential components for constructing the orthogonal basis, particularly when task-specific updates exhibit heterogeneous rank distributions.

**(ii) Dynamic Rank Adaptation**: Our sparsity-adaptive rank selection (Eq. 9) requires layer-wise sparsity measurement $\Omega_l$, computed from the full parameter deviation matrix $\boldsymbol{\Delta}_l^{(i)}$. Truncating $\boldsymbol{\Delta}_l^{(i)}$ prematurely would bias $\Omega_l$ by excluding contributions from low-magnitude parameters, thereby undermining the adaptive rank calculation. As shown in Algorithm 1, orthogonalization (Step 4) utilizes the full-rank SVD decomposition to maintain statistical fidelity.

**(iii) Outlier Weighting Integrity**: The outlier-aware weighting mechanism (Eq. 6) operates on the complete parameter deviation matrix to identify statistically significant updates. Rank selection prior to outlier detection would risk eliminating subtle yet critical features masked within lower-rank components, particularly in layers with heavy-tailed parameter distributions.

# B More Details for Related Work

## B.1 Discussion with the Alignment Tax.

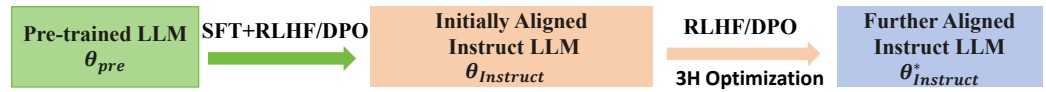

Figure 4: Illustration of Training Stage of 3H Optimization, which aims to further enhance LLMs' alignment from three perspectives based on the existing Initially Aligned LLMs.

We would like to further clarify the main difference between the 3H trade-off and the previously defined alignment tax [37, 28]. In general, the alignment tax describes the phenomenon of RLHF training leading to *the forgetting of pre-trained abilities during the first alignment stage*. However, as shown in Figure 4, we mainly focus on how we can further *enhance the 3H-related abilities of the existing already-aligned model during the second or subsequent stages*. The trade-off mainly comes from the conflict of different alignment objects without dealing with the pre-trained knowledge. Take the Llama3 series for example, alignment tax mainly analyzes the pre-trained ability degradation on the SFT version of the Base LLM (e.g., train the Llama-3-8B on the Ultrachat) while performing DPO training, which refers to the **green arrow** of the Figure 4. However, in this paper, we mainly focus on how can we further enhance the 3H-related abilities of the existing already aligned model (e.g. Llama3-8B-Instruct) during the second or subsequent alignment stages (**orange arrow** of the Figure 4), which can meet more strict demands for specific applications.

## B.2 Discussion with the Other Model Merging Methods

To further distinguish our work from previous ones and strengthen our contribution, we provide more detailed discussions about the other model merging methods.

**MOE-based merging works need additional input data to train the router**: These works, such as SMILES [86], Free-Merging [87], and Twin-Merging [87], aim to balance the performance and deployment costs through modular expertise identification and integration adapted to the input data, which is not designed for 3H optimization in LLM alignment. Recently, we have noticed a concurrent

MOE-fusion work called H3 fusion [25] related to our theme. It includes three main steps:(i) Adopt the instruction tuning and summarization fusion as two modern ensemble learning in the context of helpful-harmless-honest (H3) alignment (ii) **Merge** the aligned model weights with an expert router **according to the type of input** instruction and dynamically select a subset of experts. (iii) Utilize the gating loss and regularization terms to enhance performance. But our work mainly focuses on how we can address the conflict issue for 3H optimization to construct a multi-object aligned LLM rather than dynamically adapting to the input data. Simultaneously, considering that the constraints of data availability and data leak will limit the generalization of existing merging methods for LLMs, in the paper we mainly adopt the well-known and latest **training-free and data-free** merging strategies for dense LLM, while H3 fusion needs the data for training and only utilizes the merging techniques for efficiently adapting to the input data. Thus, **H3 fusion is indeed different from our work from the perspective of problem and technique contributions.**

**Other training-based merging works need additional data for test-time adaptation optimization:** These works, such as Adamerging [88], AIM-merging[89], Sense-merging[90], Adarank [62], DAM [34], utilize the test-time-adaptation techniques to search for the optimal merging coefficient or prune the rank [91]. Their effectiveness depends heavily on the provided test data. But for 3H optimization, curating high-quality preference data that meets the demand of helpfulness, harmlessness, and honesty simultaneously is difficult due to the complex collective and conflict relationships as stated above. We also need to consider the data mixture problems during test-time adaptation optimization. In this case, we can't compare data mixture and model merging methods for 3H optimization fairly. That's why we only compare the training-free model merging methods in our experimental parts.

### B.3 Discussion with the Outlier-Based Sparse LLM Works

To further distinguish our work from previous ones and strengthen our contribution, we provide more detailed discussions about the outlier-based sparse LLM works [92, 93].

Many works investigate the outlier weight in transformer [94, 95] and propose to prune LLM assisted by input activations [82, 83] or sample layer-wise weight during fine-tuning [96]. From the perspective of outlier weight source, the outlier weight updates we addressed are due to the preference noise accumulation while merging different aligned LLMs, which is a special problem for merging for multi-objective alignment. From the perspective of the status of the training process, previous outlier-based sparsity LLM works are only constrained to the parameters of one LLM [84, 85], while we should additionally consider the parameters conflict while merging different LLMs in the process, rather than a post-hoc process on a well-merged LLM. That's why we first perform SVD analysis to separate task-specfic parameters and only adopt outlier-weighting on the singular value.

## C    More Details for Experiments

### C.1    The Training Details for Model Constructions and Baselines

**Training hyperparameters for model constructions:** following SimPO [72], based on Llama-3-8B-Instruct and Mistral-7B-Instruct-V2, we conduct preference optimization adopting the fixed batch size 128 for 1 epoch training with the Adam optimizer. We set the max sequence length to 4096 and apply a cosine learning rate schedule with 10 percent warmup steps for each dataset. Specially, we adjust $\beta \in [0.1, 0.5, 1.0, 2.0]$ and learning rate $lr \in [3e-7, 5e-7]$ for model constructions and report the best individual training models corresponding to different annotation dimensions.

**The Implementation of Baselines:** For Heuristic data mixture methods, we control the ratio between Honesty&Harmlessness and Helpfulness to 1/5,1/10, and 1/20 by default and report the best average score (usually 1/10 according to our experiments). For ArmoRM, we follow the process of SimPO [72] to achieve refined full mixture data. For hummer [15], we refine the alignment dimension conflict (ADC) among preference datasets, leveraging the powerful ability of AI feedback (e.g., GPT4) as the paper stated. For the full mixture datasets of Table 1, we control the ADC lower than 20 percent.

**Computation environment:** All of our experiments in this paper were conducted on 16×A100 GPUs based on the Llama-Factory [97],MergeKit [75] and fusion bench [98].

**Reproducibility:** We have made significant efforts to ensure the reproducibility of our work. Upon acceptance, we will release all of the trained models and the complete training and testing code to

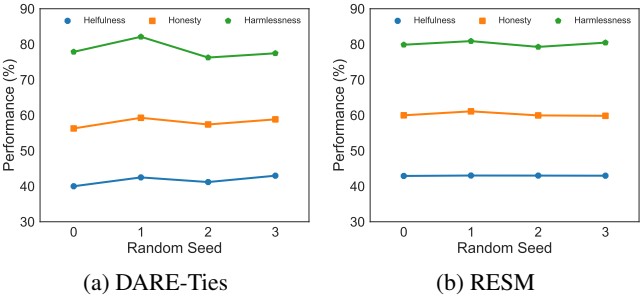

|              (a) DARE-Ties                    (b) RESM              |

Figure 5: Comparisons between the random sparsification strategy (e.g.DARE-Ties) and SVD-based strategy (RESM on Mistral under static optimization settings adopting different seeds. RESM can achieve more stable results than random sparsification methods.

facilitate the full reproducibility of our results. We are committed to advancing this work and will provide updates on its accessibility in the future.

## C.2 The Evaluation Details for the Judged Models

We provide detailed descriptions for the evaluation that needs the judged models. For MT-Bench, we report scores following its evaluation protocol to grade single answers from 1 to 10 scores assisted by GPT4. For HaluEval-Wild, given prompts to our trained model, we utilize the judged model to check whether the output of our trained model is a hallucination or not and then calculate the no hallucination rate. Similarly, we utilize the prompts from SaladBench and OR-Bench to instruct our trained models and then let the judged models check whether the replies of our trained models are safe/unsafe or refusal/answer. Based on the check results, we can naturally calculate the safe score and refusal score by counting all results. The detailed descriptions of the evaluation can be shown in Table 9. More details can be shown in the original paper.

Table 9: Evaluation details corresponding judge models, scoring types, and metrics.

| Evaluation Datasets | Examples | Judge Models | Scoring Type | Metrics |
|---|---|---|---|---|
| MT-Bench [78] | 80 | GPT-4 | Single Answer Grade | Rating of 1-10 |
| HaluEval-Wild [79] | 500 | GPT4 | Classify & Calculate Ratio | Rating of 0-100 |
| SaladBench [80] | 1817 | MD-Judge-V0.2 | Classify & Calculate Ratio | Rating of 0-100 |
| OR-Bench [81] | 1319 | GPT4-o | Classify & Calculate Ratio | Rating of 0-100 |

## C.3 More Experiments under the Continual DPO Training Settings

As shown in Table 11, we provide additional results under the continual training settings. Through comparison results between different training stages, we can observe that the honesty, helpfulness,

Table 10: 3H Results on Llama3 Under Continuous Optimization Setting. The normalized gain metric is the average value of relative gain for each dimension compared with the results of Llama3-8B-Instruct.

| Methods | Helpfulness | | | | | | | | Honesty | Harmlessness | | Helpful_Avg | Honest_Avg | Harmless_Avg | Norm_Gain |
|---|---|---|---|---|---|---|---|---|---|---|---|---|---|---|---|
| | Math | GSM8K | ARC-E | ARC-C | MMLU | MBPP_Plus | HumanEval_Plus | MT-Bench | HaluEval_Wild | Salad_Bench | OR-Bench | | | | |
| **Llama3-8B-Instruct** | 28.08 | 78.09 | 93.65 | 82.03 | 68.20 | 58.99 | 53.05 | 8.25 | 53.50 | 91.16 | 26.97 | 58.79 | 53.50 | 59.07 | — |
| Continual DPO Training Stage1 | 29.60 | 77.63 | 93.47 | 82.71 | 68.33 | 59.79 | 59.15 | 8.18 | 56.00 | 90.86 | 39.80 | 59.86↑1.82% | 56.00↑4.67% | 65.33↑10.60% | +5.70% |
| Continual DPO Training Stage2 | 28.74 | 74.60 | 94.00 | 83.05 | 68.41 | 51.59 | 56.10 | 8.25 | 52.20 | 90.55 | 77.95 | 58.09↑1.19% | 52.20↑2.43% | 84.25↑42.59% | +13.00% |
| Continual DPO Training Stage3 | 28.66 | 76.12 | 93.05 | 82.83 | 68.40 | 54.57 | 56.10 | 8.03 | 53.20 | 90.63 | 71.61 | 58.72↓0.12% | 53.20↓0.56% | 81.12↑37.30% | +12.21% |
| Weight Average | 29.78 | 79.82 | 93.65 | 82.37 | 68.40 | 58.47 | 53.65 | 8.03 | 53.20 | 89.58 | 62.66 | 59.27↑0.82% | 53.20↓0.56% | 76.12↑28.83% | +9.70% |
| Rewarded Soup | 29.40 | 79.76 | 93.65 | 82.37 | 68.48 | 58.47 | 54.88 | 8.15 | 54.20 | 89.33 | 62.75 | 59.40↑1.04% | 54.20↑1.31% | 76.04↑28.69% | +10.01% |
| Model Stock | 28.42 | 79.15 | 93.65 | 82.37 | 68.30 | 60.05 | 53.05 | 8.25 | 50.60 | 91.27 | 28.96 | 59.16↑0.63% | 50.60↓5.42% | 60.12↑1.78% | -1.00% |
| Task Arithmetic | 28.72 | 73.16 | 92.95 | 83.05 | 68.32 | 52.11 | 46.34 | 8.52 | 51.60 | 86.07 | 84.97 | 56.65↓3.64% | 51.60↓3.55% | 85.52↑44.74% | +12.52% |
| Ties | 29.18 | 76.50 | 93.65 | 83.39 | 68.61 | 56.35 | 43.78 | 7.71 | 52.80 | 87.55 | 78.59 | 57.40↓2.36% | 52.80↓1.31% | 83.07↑40.60% | +12.31% |
| DARE | 28.18 | 73.92 | 92.95 | 83.05 | 68.30 | 51.85 | 49.39 | 8.02 | 52.00 | 85.76 | 85.75 | 56.96↓3.11% | 52.00↓2.80% | 85.76↑45.15% | +13.08% |
| DARE Ties | 29.48 | 78.85 | 93.65 | 82.37 | 68.43 | 59.79 | 53.66 | 7.67 | 52.40 | 89.46 | 71.38 | 59.24↑0.77% | 52.40↓2.06% | 80.42↑36.11% | +11.61% |
| DELLA | 27.68 | 71.19 | 93.12 | 83.05 | 68.31 | 48.15 | 46.34 | 8.15 | 51.80 | 86.58 | 87.11 | 55.75↓5.17% | 51.80↓3.18% | 86.85↑47.00% | +12.89% |
| DELLA Ties | 28.94 | 72.18 | 93.47 | 82.71 | 68.41 | 53.97 | 47.56 | 8.21 | 52.20 | 87.24 | 84.38 | 56.93↓3.16% | 52.20↓2.43% | 85.81↑45.24% | +13.22% |
| Breadcrumbs | 28.92 | 78.62 | 93.47 | 82.71 | 68.45 | 55.82 | 50.00 | 8.48 | 52.40 | 87.88 | 72.69 | 58.31↓0.82% | 52.40↓2.06% | 80.29↑35.89% | +11.00% |
| Breadcrumbs Ties | 29.79 | 78.77 | 93.65 | 83.73 | 68.37 | 57.41 | 56.10 | 8.57 | 53.40 | 88.26 | 67.64 | 59.55↑1.29% | 53.40↓0.19% | 77.95↑31.96% | +11.02% |
| TSVM | 29.86 | 78.99 | 93.65 | 83.71 | 68.37 | 58.51 | 55.40 | 8.40 | 53.80 | 88.68 | 75.14 | 59.61↑1.39% | 53.80↑0.56% | 81.79↑38.46% | +13.47% |
| **RESM(ours)** | 29.79 | 78.77 | 93.65 | 83.73 | 68.37 | 58.45 | 56.10 | 8.57 | 54.50 | 89.26 | 75.34 | 60.05↑2.14% | 54.50↑1.87% | 82.49↑39.66% | **+14.56%** |

Table 11: 3H Results on Mistral Under Continuous Optimization Setting. The normalized gain metric is the average value of relative gain for each dimension compared with the results of Mistral-7B-Instruct-V2.

| Methods | Helpfulness | | | | | | | | Honesty | Harmlessness | | Helpful_Avg | Honest_Avg | Harmless_Avg | Norm_Gain |
|---|---|---|---|---|---|---|---|---|---|---|---|---|---|---|---|
| | Math | GSM8K | ARC-E | ARC-C | MMLU | MBPP_Plus | HumanEval_Plus | MT-Bench | HaluEval_Wild | Salad_Bench(↑) | OR-Bench(↑) | | | | |
| **Mistral-7B-Instruct-V2** | 9.54 | 46.17 | 82.36 | 72.88 | 59.97 | 26.46 | 28.66 | 7.55 | 62.17 | 78.07 | 74.68 | 41.70 | 62.17 | 76.38 | — |
| Continual DPO Training Stage1 | 8.76 | 43.14 | 82.01 | 74.92 | 59.78 | 25.93 | 27.33 | 7.59 | 61.33 | 78.74 | 77.23 | 41.18$_{↓1.25\%}$ | 61.33$_{↓1.35\%}$ | 77.99$_{↑2.11\%}$ | -0.16% |
| Continual DPO Training Stage2 | 9.26 | 36.16 | 82.54 | 75.59 | 60.38 | 29.88 | 33.33 | 7.86 | 56.40 | 82.76 | 78.54 | 41.88$_{↑0.43\%}$ | 56.40$_{↓9.28\%}$ | 80.65$_{↑5.59\%}$ | -1.09% |
| Continual DPO Training Stage3 | 9.60 | 40.49 | 82.54 | 77.29 | | 26.25 | 34.15 | 7.46 | 57.40 | 80.77 | 83.16 | 42.29$_{↑1.41\%}$ | 57.40$_{↓7.67\%}$ | 81.97$_{↑7.32\%}$ | +0.35% |
| Weight Average | 10.04 | 45.72 | 82.36 | 75.25 | 61.03 | 26.46 | 31.71 | 7.56 | 59.20 | 78.02 | 81.43 | 42.52$_{↑1.97\%}$ | 59.20$_{↓4.78\%}$ | 79.73$_{↑4.38\%}$ | +0.52% |
| Rewarded Soup | 9.72 | 46.02 | 82.19 | 75.25 | 61.03 | 26.46 | 32.93 | 7.61 | 58.60 | 77.94 | 81.34 | 42.65$_{↑2.28\%}$ | 58.60$_{↓5.74\%}$ | 79.64$_{↑4.27\%}$ | +0.27% |
| Model Stock | 9.74 | 47.69 | 82.36 | 73.56 | 59.77 | 24.87 | 27.44 | 7.68 | 61.00 | 78.51 | 76.44 | 41.64$_{↓0.14\%}$ | 61.00$_{↓1.88\%}$ | 77.48$_{↑1.44\%}$ | -0.53% |
| Task Arithmetic | 9.76 | 43.06 | 82.54 | 75.93 | 61.27 | 25.66 | 32.93 | 7.46 | 57.80 | 78.32 | 82.35 | 42.33$_{↑1.51\%}$ | 57.80$_{↓7.03\%}$ | 80.34$_{↑5.18\%}$ | -0.11% |
| Ties | 10.48 | 41.55 | 84.66 | 76.27 | 61.60 | 26.19 | 30.49 | 7.46 | 53.80 | 78.99 | 85.43 | 42.34$_{↑1.53\%}$ | 53.80$_{↓13.46\%}$ | 82.21$_{↑7.64\%}$ | -1.43% |
| DARE | 10.40 | 42.99 | 85.36 | 75.93 | 61.54 | 24.60 | 33.54 | 7.54 | 56.00 | 78.81 | 85.21 | 42.74$_{↑2.49\%}$ | 56.00$_{↓9.92\%}$ | 82.01$_{↑7.37\%}$ | -0.02% |
| DARE Ties | 10.28 | 42.00 | 85.01 | 76.27 | 61.61 | 27.25 | 32.32 | 7.43 | 53.00 | 79.17 | 86.50 | 42.77$_{↑2.57\%}$ | 53.00$_{↓14.75\%}$ | 82.84$_{↑8.46\%}$ | -1.24% |
| DELLA | 10.18 | 43.14 | 84.83 | 75.25 | 61.46 | 26.46 | 31.71 | 7.58 | 55.25 | 79.35 | 86.04 | 42.58$_{↑2.11\%}$ | 55.25$_{↓11.13\%}$ | 82.70$_{↑8.28\%}$ | -0.25% |
| DELLA Ties | 10.50 | 40.18 | 85.89 | 77.97 | 61.37 | 30.16 | 30.48 | 7.30 | 54.80 | 79.90 | 87.49 | 42.98$_{↑3.07\%}$ | 54.80$_{↓11.85\%}$ | 83.70$_{↑9.58\%}$ | +0.27% |
| Breadcrumbs | 10.56 | 42.53 | 84.83 | 75.59 | 64.50 | 24.60 | 32.32 | 7.53 | 52.40 | 79.42 | 84.34 | 42.81$_{↑2.66\%}$ | 52.40$_{↓15.71\%}$ | 81.88$_{↑7.20\%}$ | -1.95% |
| Breadcrumbs Ties | 10.54 | 42.46 | 84.66 | 76.95 | 61.47 | 26.72 | 29.88 | 7.45 | 53.40 | 79.80 | 84.57 | 42.52$_{↑1.97\%}$ | 53.40$_{↓14.11\%}$ | 82.19$_{↑7.61\%}$ | -1.51% |
| TSVM | 10.52 | 41.25 | 85.28 | 77.21 | 61.57 | 29.22 | 30.48 | 7.55 | 54.95 | 79.90 | 87.49 | 42.89$_{↑2.85\%}$ | 54.95$_{↓11.61\%}$ | 83.70$_{↑9.58\%}$ | +0.27% |
| **RESM(ours)** | 10.61 | 41.26 | 85.18 | 77.96 | 61.60 | 29.91 | 31.57 | 7.65 | 59.15 | 79.94 | 87.69 | 43.25$_{↑3.72\%}$ | 59.15$_{↓4.97\%}$ | 83.82$_{↑9.75\%}$ | +2.83% |

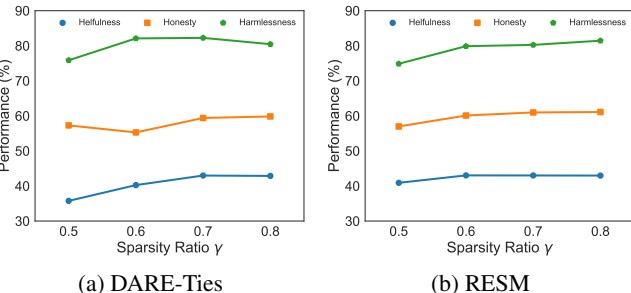

(a) DARE-Ties      (b) RESM

Figure 6: Parameter sensitive analysis concerning sparsity factor for model merging methods on Mistral under static optimization settings.

and harmlessness of LLMs are interactively enhanced due to forgetting during continual training. Moreover, model merging methods can achieve comparable results to these continual training methods without the need to consider the optimized status at a specific training stage. In other words, model merging paves a new way for continual DPO training, advocating training multiple models from the same start point and then merging them, rather than continually optimizing the model from the previous optimization.

## C.4 Hyper-Parameter Analysis

The sparsity-based strategy is closely related to the merging effect. As shown in Table 3 and Table 4, apart from the SVD-based methods, the most effective merging methods are DARE and DELLA, both of which depend on random sparsification as shown in Table 2. However, we conduct extended studies to check the robustness and stability with respect to random seed and sparsity factors. As shown in Figure 5 and Figure 6, we can observe that RESM can achieve better robust results than previous random sparsification-based methods, further verifying the effectiveness of our methods.

## D Broad Impact and Limitation

Our results demonstrate that the main improvement of RESM comes from the honest and harmless aspects. This can reflect the decrease in conflict between them, which can be defined as inter-aspect conflict reduction. But for helpfulness, RESM is still worse than data mixture methods on Llama3, and the improvement on Mistral compared with the existing merging strategy is also minimal. Though the initial goal of honest and harmless training is not designed for helpfulness, modern preference datasets inherently encode helpfulness as a baseline annotation, forcing the alignment process to optimize towards this dimension regardless of their primary target (honesty/harmlessness). This means every alignment vector can represent helpfulness and one or more other dimensions' optimization directions, which may lead to conflict between alignment vectors only from the helpful dimension (e.g. code and commonsense QA abilities for LLM), which can be defined as intra-dimension conflict. This phenomenon necessitates a hierarchical conflict resolution framework to improve model merging for 3H optimization, considering these two categories of conflicts simultaneously. Moreover, deploying

model merging methods for other trustworthy concerns [99–103] across more diverse circumstances [104–107, 106] should also be considered.

