# OpenReview forum: "Mix Data or Merge Models? Balancing the Helpfulness, Honesty, and Harmlessness of Large Language Model via Model Merging"
_NeurIPS.cc/2025/Conference — NeurIPS 2025 poster_

### Official Review · Reviewer_UavS · 2025-07-01

**Clarity:** 3
**Significance:** 3
**Originality:** 3
**Rating:** 4
**Confidence:** 3

**Summary:**

This paper systematically compares data mixture and model merging approaches for aligning large language models (LLMs) across Helpfulness, Honesty, and Harmlessness (3H) dimensions. The authors establish the first comprehensive 3H alignment benchmark and reveal that model merging, with proper conflict mitigation, can outperform data mixture strategies. To further enhance alignment, they propose a novel Reweighting-Enhanced Task Singular Vector Merging (RESM) method, which addresses preference noise and layer-wise sparsity through outlier-aware weighting and adaptive rank selection. Extensive experiments demonstrate the effectiveness of RESM in achieving balanced 3H alignment for LLMs.

**Questions:**

1）Mistral-7B is no longer state-of-the-art. To strengthen generality, please add experiments using a stronger baseline (e.g., Qwen-3-7B).
2）In Table 3, individual "Harmlessness" training yields lower Harmlessness scores (60.14) than RESM (79.60). Why does merging non-harmlessness-specialized models (e.g., Helpfulness-trained) improve this metric beyond dedicated training?
3）Your method's performance improvement seems to mainly take effect on honesty and harmlessness, but the evaluation datasets corresponding to these two indicators are significantly fewer than those for helpfulness. This only seems to demonstrate the effectiveness of your method for the three evaluation datasets under honesty and harmlessness, rather than for balancing the 3H (honesty, harmlessness, helpfulness). Please supplement evaluation datasets for honesty and harmlessness respectively (5 for honesty and 5 for harmlessness).

**Ethical Concerns:**

["NO or VERY MINOR ethics concerns only"]

**Final Justification:**

The author's response addressed my concerns and I raised my score to 4.

**Limitations:**

yes

**Quality:**

2

**Strengths And Weaknesses:**

Strengths:
The paper addresses the critical and underexplored problem of achieving balanced alignment for LLMs across Helpfulness, Honesty, and Harmlessness (3H), which is highly relevant to the current trend of safe and responsible AI development.
The authors establish the first comprehensive benchmark for comparing data mixture and model merging approaches for 3H alignment, covering 15 representative methods, 10 preference datasets, and 2 mainstream LLM families. This provides valuable experimental evidence and standard evaluation criteria for the field.
The proposed Reweighting-Enhanced Task Singular Vector Merging (RESM) method introduces a principled way to address preference noise accumulation and layer-wise sparsity heterogeneity in model merging, which are critical yet often overlooked challenges.
Weaknesses:
While RESM introduces useful improvements, it builds heavily on existing SVD-based merging frameworks (e.g., TSVM). Also, while the empirical results are strong, the theoretical foundations for why RESM consistently improves 3H alignment remain somewhat heuristic. More formal analysis or theoretical guarantees would strengthen the work.
The paper lacks discussion on the computational overhead and scalability of RESM. Meanwhile, the models used are not the most state-of-the-art ones, and the evaluation metrics for Honesty and Harmlessness are too single-dimensional. (See Q1-Q2)

---

> ### Author Rebuttal · Authors · 2025-07-31
>
> Thanks for your valuable suggestions! We are glad to address your concerns one by one.
>
> **Q1: More experiments on the latest models**
>
> Due to the time limit, we will add the experiments later.
>
> **Q2: More formal analysis or theoretical guarantees would strengthen the work.**
>
> Thanks for your suggestions. To verify the effectiveness, in the Appendix, we provide the interpretation of dual objectives for outlier weighting from cross-model consensus and intra-model saliency, conflict probability bound for adaptive rank selection,  and the order of orthogonalization and rank selection. We will further strengthen our work referring to your suggestions.
>
> **Q3: More balanced Evaluation. This only seems to demonstrate the effectiveness of your method for the three evaluation datasets under honesty and harmlessness, rather than for balancing the 3H (honesty, harmlessness, helpfulness). Please supplement evaluation datasets for honesty and harmlessness respectively (5 for honesty and 5 for harmlessness).**
>
> Exactly, the data sources of harmlessness and honesty are also diverse enough, not limited to the shown absolute number of datasets. For example, the harmlessness evaluation saladbench includes more than 9 datasets (HH-harmless, HH-red-team, Advbench, Multilingual, Do-Not-Answer, ToxicChat, Do Anything Now, GPTFuzzer). We have tried our best to make the evaluation extensive and reasonable.
>
> Moreover, in terms of the dataset quantity, the general abilities evaluation datasets themselves are superior to the trustworthy capability evaluation datasets. That's why we propose to aggregate the results for each dimension individually for a fair and reasonable evaluation
>
> **Q4: The paper lacks discussion on the computational overhead and scalability of RESM.**

---

> > ### Comment · Reviewer_UavS · 2025-08-07
> >
> > The author's response addressed my concerns and I will consider raising my score.

---

> > > ### Author Response · Authors · 2025-08-07
> > > **Reply to Reviewer UavS**
> > >
> > > Thanks again for your insightful feedback! We will carefully organize the content of the rebuttal in accordance with your suggestions.

---

> ### Author Response · Authors · 2025-08-01
> **Updates for Rebuttal**
>
> Due to the time limit in the previous phase, we would like to provide additional experiments to further address your questions.
>
> **Q1: More experiments on the latest models**
>
> Thanks for your questions. We would like to clarify that our 3H DPO training executes on the already aligned Instruct models, which corresponds to the most commonly used preference optimization training setting (same as SimPO[1], adopting llama3 and Mistral model families). Thus, **our submission version follows these settings for better and fairer comparisons** of different merging algorithms.
>
> To further address your concerns, we **provide the Qwen2.5-3B-Instruct experiments** considering the computational resource constraint. The detailed experimental results can be shown as follows:
>
> |                               | Helpfulness  | Honesty      | Harmlessness  | Norm_Gain |
> |-------------------------------|--------------|--------------|---------------|-----------|
> |      Qwen2.5-3B-Instruct      |     75.41    |     67.42    |     71.25     |     —     |
> |        Helpful Training       | 78.95(+4.7%) | 68.42(+1.5%) |  72.45(+1.7%) |   +2.63%  |
> |        Honest Training        | 76.85(+1.9%) | 70.42(+4.5%) |  68.45(-3.9%) |   +0.83%  |
> |       Harmless Training       | 77.44(+2.7%) | 66.98(-0.7%) |  74.15(+4.1%) |   +2.03%  |
> |  Helpful and Honest Training  | 79.95(+6.0%) | 71.42(+5.9%) |  72.15(+1.3%) |   +4.40%  |
> | Helpful and Harmless Training | 79.34(+5.2%) | 68.45(+1.5%) | 78.45(+10.1%) |   +5.60%  |
> |   3H Training (Heurisistic)   | 78.45(+4.0%) | 68.58(+1.7%) |  72.42(+1.6%) |   +2.43%  |
> |      3H Training (Armo）      | 79.35(+5.2%) | 69.51(+3.1%) |  72.58(+1.9%) |   +3.40%  |
> |      3H Training (Hummer)     | 80.11(+6.2%) | 70.15(+4.0%) |  74.23(+4.2%) |   +4.80%  |
> |         Weight Average        | 79.80(+5.8%) | 69.72(+3.4%) |  73.85(+3.7%) |   +4.30%  |
> |         Rewarded Soup         | 79.64(+5.6%) | 70.96(+5.2%) |  75.08(+5.4%) |   +5.40%  |
> |          Model Stock          | 78.02(+3.5%) | 68.24(+1.2%) |  72.00(+1.1%) |   +1.93%  |
> |         Task Arithemic        | 78.02(+3.5%) | 69.61(+3.2%) |  72.33(+1.5%) |   +2.73%  |
> |          Ties Merging         | 78.42(+4.0%) | 69.26(+2.7%) |  73.71(+3.5%) |   +3.40%  |
> |      Dare Task Arithemic      | 79.42(+5.3%) | 70.41(+4.4%) |  74.00(+3.9%) |   +4.53%  |
> |       Dare Ties Merging       | 79.85(+5.9%) | 70.85(+5.1%) |  75.23(+5.6%) |   +5.53%  |
> |          Breadcrumbs          | 78.95(+4.7%) | 69.58(+3.2%) |  73.45(+3.1%) |   +3.67%  |
> |        Breadcrumbs_Ties       | 79.35(+5.2%) | 70.15(+4.0%) |  74.58(+4.7%) |   +4.63%  |
> |             Della             | 79.11(+4.9%) | 69.92(+3.7%) |  74.15(+4.1%) |   +4.23%  |
> |           Della_Ties          | 79.55(+5.5%) | 70.42(+4.5%) |  75.45(+5.9%) |   +5.30%  |
> |              TSVM             | 78.58(+4.2%) | 69.35(+2.9%) |  73.42(+3.0%) |   +3.37%  |
> |         **RESM(Ours)**        | 80.45(+6.7%) | 71.42(+5.9%) |  76.15(+6.9%) |   +6.50%  |
>
> From the results, we can observe that our method still achieves better results than previous data mixture and model merging methods, which can further verify the effectiveness of our method
>
>
> **Q4: The paper lacks discussion on the computational overhead and scalability of RESM.**
>
> Thanks for your good advice. We provide a detailed merging time analysis between two layer-wise merging algorithms(TSVM and RESM) in terms of different numbers of models as follows. Our experiments are conducted on 32xA100 devices. From the results, we can find that training-free model merging methods consistently don't need too much time for merge executions. Compared with TSVM, the additional time cost of RESM can be accepted in a way.
>
> |           | TSVM     | RESM    |
> |-----------|----------|---------|
> | 3 models  | 2 min    | 2.2 min |
> | 5 models  | 2.5 min  | 2.8 min |
> | 10 models | 3 min    | 3.3 min |
> | 20 models | 4.5 min  | 5 min   |
>
> **Q5: In Table 3, individual "Harmlessness" training yields lower Harmlessness scores (60.14) than RESM (79.60). Why does merging non-harmlessness-specialized models (e.g., Helpfulness-trained) improve this metric beyond dedicated training?**
>
> Thanks for your questions. From the annotation data stastics of DPO training of Table1, we can find that harmness training only utilize the UltraSafety datasets, but RESN merges five aligned models (including Helpfulness, Harmlessness,Honesty,Helpfulness & Honesty and Helpfulness & Harmlessness). The Helpfulness & Harmlessness also contribute to the gain of harmlessness metric.
>
> We hope these statements can address your concerns. Thank you for your time and valuable feedback on our work.
>
> [1] NeurIPS2024, Simpo: Simple preference optimization with a reference-free reward

---

> ### Author Response · Authors · 2025-08-06
> **Kind Reminder to Reviewer UavS**
>
> Dear Reviewer UavS,
>
> Thank you for your time and valuable feedback on our work. As the reviewer reply period is coming to the end, we kindly encourage you to share any remaining questions or concerns at your earliest convenience, as we will not be able to respond after this period.
>
> Your insights are greatly appreciated and have significantly contributed to improving our paper. Of course, we sincerely hope you can reconsider the score if the mentioned problems have been solved.
>
> Best regards,
>
> The Authors

---

### Official Review · Reviewer_jwRS · 2025-07-02

**Clarity:** 3
**Significance:** 3
**Originality:** 3
**Rating:** 4
**Confidence:** 2

**Summary:**

This paper aims to solve the inherent conflicts of large language models (LLMs) in the alignment of 3H (helpfulness, honesty, harmless) multiple objectives. To this end, it establishes a comprehensive benchmark and systematically compares 15 representative methods upon this benchmark. Additionally, the authors figure out two problems in the existing merging methods and propose a novel RESM (Reweighting-Enhanced task Singular Merging) algorithm to overcome them. Extensive experiments have shown that RESM is significantly superior to existing data mixing and model merging methods in achieving 3H balance.

**Questions:**

N.A.

**Ethical Concerns:**

["NO or VERY MINOR ethics concerns only"]

**Final Justification:**

The author's reply addresses my concerns about this work. And I believe my current score has shown my positive attitude on this paper and will keep it.

**Limitations:**

yes

**Paper Formatting Concerns:**

N.A.

**Quality:**

3

**Strengths And Weaknesses:**

* Strengths
1. This work conducts a systematic review on existing methods about 3H-alignment, and designs comprehensive benchmark to evaluate these methods from multiple dimensions. The extensive experiment results offer valuable insights to the community (e.g.,  a phased "specialize-then-merge" paradigm may be a more effective path to multi-objective alignment than a monolithic data-mixing approach).
2. This paper proposes a new algorithm named RESM to overcome two unsolved limitations, i.e., Preference Noise Accumulation and Layer-wise Sparsity Mismatch. The design constitutes a high-quality technical contribution and is well validated by the experiments.

* Weakness
1. While assembling public resources into a structured benchmark is a valuable engineering effort, the most selected datasets in the benchmark is widely used in the community (such as MATH, GSM8k, MMLU, HumanEval, etc.). I'm not so sure whether such common combinations deserve a critical contribution of this work.
2. The LLM landscape evolves rapidly. Although Llama-3 and Mistral are highly representative, the experiments lack validation on more recent and capable open-source models (e.g., the Qwen-2.5, DeepSeek-R1-distilled models). Evaluating existing methods and the new-proposed RESM to these cutting-edge models would more powerfully demonstrate the generalizability of this benchmark and brings more realtime insights.

---

> ### Author Response · Authors · 2025-08-01
> **Updates for Rebuttal**
>
> Due to the time limit in the previous phase, we would like to provide additional experiments to further address your questions.
>
> **Q1: More experiments on the latest models(e.g Qwen2.5)**
>
> Thanks for your questions. We would like to clarify that our 3H DPO training executes on the already aligned Instruct models, which corresponds to the most commonly used preference optimization training setting (same as SimPO[1], adopting llama3 and Mistral model families). Thus, **our submission version follows these settings for better and fairer comparisons** of different merging algorithms.
>
> To further address your concerns, we **provide the Qwen2.5-3B-Instruct experiments** considering the computational resource constraint. The detailed experimental results can be shown as follows:
>
> |                               | Helpfulness  | Honesty      | Harmlessness  | Norm_Gain |
> |-------------------------------|--------------|--------------|---------------|-----------|
> |      Qwen2.5-3B-Instruct      |     75.41    |     67.42    |     71.25     |     —     |
> |        Helpful Training       | 78.95(+4.7%) | 68.42(+1.5%) |  72.45(+1.7%) |   +2.63%  |
> |        Honest Training        | 76.85(+1.9%) | 70.42(+4.5%) |  68.45(-3.9%) |   +0.83%  |
> |       Harmless Training       | 77.44(+2.7%) | 66.98(-0.7%) |  74.15(+4.1%) |   +2.03%  |
> |  Helpful and Honest Training  | 79.95(+6.0%) | 71.42(+5.9%) |  72.15(+1.3%) |   +4.40%  |
> | Helpful and Harmless Training | 79.34(+5.2%) | 68.45(+1.5%) | 78.45(+10.1%) |   +5.60%  |
> |   3H Training (Heurisistic)   | 78.45(+4.0%) | 68.58(+1.7%) |  72.42(+1.6%) |   +2.43%  |
> |      3H Training (Armo）      | 79.35(+5.2%) | 69.51(+3.1%) |  72.58(+1.9%) |   +3.40%  |
> |      3H Training (Hummer)     | 80.11(+6.2%) | 70.15(+4.0%) |  74.23(+4.2%) |   +4.80%  |
> |         Weight Average        | 79.80(+5.8%) | 69.72(+3.4%) |  73.85(+3.7%) |   +4.30%  |
> |         Rewarded Soup         | 79.64(+5.6%) | 70.96(+5.2%) |  75.08(+5.4%) |   +5.40%  |
> |          Model Stock          | 78.02(+3.5%) | 68.24(+1.2%) |  72.00(+1.1%) |   +1.93%  |
> |         Task Arithemic        | 78.02(+3.5%) | 69.61(+3.2%) |  72.33(+1.5%) |   +2.73%  |
> |          Ties Merging         | 78.42(+4.0%) | 69.26(+2.7%) |  73.71(+3.5%) |   +3.40%  |
> |      Dare Task Arithemic      | 79.42(+5.3%) | 70.41(+4.4%) |  74.00(+3.9%) |   +4.53%  |
> |       Dare Ties Merging       | 79.85(+5.9%) | 70.85(+5.1%) |  75.23(+5.6%) |   +5.53%  |
> |          Breadcrumbs          | 78.95(+4.7%) | 69.58(+3.2%) |  73.45(+3.1%) |   +3.67%  |
> |        Breadcrumbs_Ties       | 79.35(+5.2%) | 70.15(+4.0%) |  74.58(+4.7%) |   +4.63%  |
> |             Della             | 79.11(+4.9%) | 69.92(+3.7%) |  74.15(+4.1%) |   +4.23%  |
> |           Della_Ties          | 79.55(+5.5%) | 70.42(+4.5%) |  75.45(+5.9%) |   +5.30%  |
> |              TSVM             | 78.58(+4.2%) | 69.35(+2.9%) |  73.42(+3.0%) |   +3.37%  |
> |         **RESM(Ours)**        | 80.45(+6.7%) | 71.42(+5.9%) |  76.15(+6.9%) |   +6.50%  |
>
> From the results, we can observe that our method still achieves better results than previous data mixture and model merging methods, which can further verify the effectiveness of our method.
>
> **Q2: While assembling public resources into a structured benchmark is a valuable engineering effort, the most selected datasets in the benchmark is widely used in the community (such as MATH, GSM8k, MMLU, HumanEval, etc.). I'm not so sure whether such common combinations deserve a critical contribution of this work.**
>
> Thanks for your questions. Exactly, in our experiments, we have tried our best to ensure a fair and reasonable evaluation through aggregating the results for each dimension individually. The detailed experimental setting can be also shown in Appendix. We mainly focus on the 3H metric for LLM post-training. More detailed discussions and investigation (such as conflict among different helpful tasks) can be shown in Appendix Limitation and Broad conflict.
>
> We clarify that every alignment vector can represent helpfulness and one or more other dimensions' optimization directions, which may lead to conflict between alignment vectors only from the helpful dimension (e.g. code and commonsense QA abilities for LLM), which can be defined as intra-dimension conflict. This phenomenon necessitates a hierarchical conflict resolution framework to improve model merging for 3H optimization, considering these two categories of conflicts simultaneously.
>
> We hope these statements can address your concerns. Thank you for your time and valuable feedback on our work.

---

> > ### Comment · Reviewer_jwRS · 2025-08-09
> >
> > Thanks for your reply. My concerns have been addressed.
> >
> > I believe my current score has shown my positive attitude on this paper and will keep it.

---

> > > ### Author Response · Authors · 2025-08-09
> > > **Reply to Reviewer jwRS**
> > >
> > > We are glad to hear that your concerns have been addressed! Thanks for your advice and questions again. We will carefully polish our paper, based on your reviews.

---

> ### Author Response · Authors · 2025-08-06
> **Kind Reminder to Reviewer  jwRS**
>
> Dear Reviewer jwRS,
>
> Thank you for your time and valuable feedback on our work. As the reviewer reply period is coming to the end, we kindly encourage you to share any remaining questions or concerns at your earliest convenience, as we will not be able to respond after this period.
>
> Your insights are greatly appreciated and have significantly contributed to improving our paper. Of course, we sincerely hope you can reconsider the score if the mentioned problems have been solved.
>
> Best regards,
>
> The Authors

---

### Official Review · Reviewer_HsEK · 2025-07-03

**Clarity:** 3
**Significance:** 1
**Originality:** 3
**Rating:** 3
**Confidence:** 3

**Summary:**

This paper compares data mixture and model merging for aligning LLMs on Helpfulness, Honesty, and Harmlessness. It proposes RESM, a model merging method using outlier-aware weighting and adaptive rank selection. RESM outperforms prior methods in balancing 3H alignment, showing strong results across static and continual training settings.

**Questions:**

Please see weaknesses section.

**Ethical Concerns:**

["NO or VERY MINOR ethics concerns only"]

**Final Justification:**

My core concerns regarding the uniqueness of 3H alignment and the proposed model merging method remain. I'm not convinced that the task conflicts shown are fundamentally different from those in other multi-task scenarios.

The reported improvements are mostly within the margin of random fluctuation and seem to be driven primarily by the OR-Bench results. This level of performance gain is not sufficient to prove the method's effectiveness.

**Limitations:**

- The motivation of this paper is insufficient.

- The proposed method appears to be a mere aggregation of components, lacking clear connections between them.

**Quality:**

2

**Strengths And Weaknesses:**

#### **Weaknesses**

- The proposed method is presented with considerable complexity, but it seems to be primarily a combination of existing techniques. Moreover, the relationships between the different submodules are not clearly articulated.

- There is insufficient empirical motivation to support the claim that traditional merging method performs poorly under 3H multi-objective conflicts. The experimental results appear similar to those in non-3H settings, suggesting that the so-called 3H conflict might merely reflect typical parameter space conflicts in multi-model merging scenarios.

- The conclusion that merging datasets is inferior to merging models seems rather obvious, and I did not observe any novel insights specific to the 3H scenario.

#### **Strengths**

- The writing is fluent and clear.

- The experimental section is thorough, covering a wide range of datasets and baselines.

---

> ### Author Rebuttal · Authors · 2025-07-31
>
> Thanks for your valuable suggestions! We are glad to address your concerns one by one.
>
> **Q1: Motivation. There is insufficient empirical motivation to support the claim that traditional merging method performs poorly under 3H multi-objective conflicts. The experimental results appear similar to those in non-3H settings, suggesting that the so-called 3H conflict might merely reflect typical parameter space conflicts in multi-model merging scenarios.**
>
> Thanks for your questions. We would like to distinguish 3H conflict with typical parameter space conflicts in previous multi-model merging scenarios as follows:
>
> **From the perspective of problem:**
>
> (i) We would like to clarify that we are the first to systematically scale the data mixture and model merging strategies for LLM 3H optimization.The detailed experimental results of LLMs across different human preference can contribute to large-scale industrial application compared with traditional multi-model merging methods;
>
> (ii) Exactly, traditional multi-model merging scenarios mainly assume we have different task-specific models for merging, which only focus on  the task conflict. But we not only consider the parameter conflict and but also consider the fine-grained alignment promotion simultaneously. As shown in Figure 3, we observe that there exist different collaborative and conflict relationships in terms of 3H objectives for LLM alignment. These findings strengthen the understanding of LLM alignment. These contents can be shown in lines 259-256.
>
> **From the perspective of technique:**
>
> (i) Due to the demand that we need to simultaneously consider the fine-grained alignment promotion and conflict, thus we shouldn't directly follow the traditional  multi-model merging methods, such as task-vector based methods(e.g. ties and dare) and Linear interpolation methods(e.g.rewarded soup). That's because the advancement of multi-model merging methods have the same assumption of separating task-common and task-specific parameter to cope with conflict. For 3H optimization of LLM alignment, the investigation analysis of Figure 6 and Broad Impact and Limitation shows that LLM alignment is a more complicated problem, we can't directly succeed this assumption, that's why we adopt the layer-wise SVD composition for 3H merging.
>
> (ii) There exists some specific problems for 3H alignment during model merging compared with traditional multi-model merging, as shown in Motivation Section 3.2, we first analyse the preference noise accumulation when we increase the number of aligned models through adjust alignment objectives and random seed. Then, we point out that conventional multi-model merging methods employ a uniform rank selection threshold $k$ across all layers, failing to account for layer-specific sparsity patterns and parameter importance heterogeneity in LLMs. These contents can be shown in lines 139-163.
>
> **Q2: Technique Details. The proposed method appears to be a mere aggregation of components, lacking clear connections between them.**
>
> As show in Eq. 4, the outlier weighting is operated on singular vector for orthogonalization  and the rank-selection is operated on the layer-wise.  **Both of them collectively contribute to the optimal subspace alignment and information preservation during 3H optimization.** The detailed discussions between outlier-based orthogonalization and rank selection can be shown in Appendix C.3.
>
> **The design of outlier weighting is to preventing dominance for 3H optimization:** For noise compression, we filter out low-magnitude fluctuations that predominantly encode noise, forcing the singular vectors to align with statistically significant task features. ensuring balanced contributions from diverse tasks and preventing dominance by high-magnitude updates that may obscure subtle yet critical features. For task equilibrium: The layer-wise aggregation outlier weights are globally normalized across all models, ensuring balanced contributions from diverse tasks and preventing dominance by high-magnitude updates that may obscure subtle yet critical features. These contents can be shown in lines 182-188.
>
> **The design of rank selection is adapted to the LLM sparsity:** For dense layers, where parameter updates demonstrate relatively uniform distributions with predominantly small adjustments, employing lower-rank approximations proves effective for noise suppression while preserving principal components. However, this necessitates careful determination of the optimal rank selection threshold to balance between information retention and noise elimination. Conversely, sparse layers display concentrated parameter updates along a few dominant directions, potentially containing critical outlier components. Here, maintaining a higher rank becomes essential to ensure the preservation of these salient directional features, thereby preventing substantial information loss through excessive rank truncation.
>
> **Apart from achieving the outlier-weighting orthogonalization singular vector, by controling the level of preserving dominant singular directions in sparse and dense layers , we reduce overlaps between task-specific parameters, decoupling interference-prone optimization trajectories.**
>
>
> **Q3: The conclusion that merging datasets is inferior to merging models seems rather obvious, and I did not observe any novel insights specific to the 3H scenario.**
>
> Thanks for your questions. Exactly, we can address the alignment conflict both from the data and parameter perspective.
>
> (i) Notably, not all merging methods achieve better results than model merging methods. For example, in Table 3, only 3/13 merging methods achieve better results than data mixture strategies.
>
> (ii) Apart from experimental results, we aim to provide insights to analyse the 3H alignment collaborative and conflict relationships (lines 259-266) and the mechanism​ of merging for 3H merging. We provide the detailed analysis when merging is better and the factors that influence merging effect. These conclusions can contribute to the community to have a better understanding of the merging and data mixture strategies for LLM alignment.These contents can be shown in lines 286-301

---

> ### Comment · Reviewer_HsEK · 2025-08-06
>
> Thanks to the author for the detailed response. However, my concerns have not been fully addressed.
>
> I am still not convinced about the fundamental distinction between 3H alignment and other downstream tasks. In the motivational experiment presented in Fig. 3, performance on honesty improves after training on in-domain data and degrades otherwise, with helpfulness showing a similar pattern. This phenomenon appears to be a typical manifestation of task conflict, and it is not clear how this differs from conflicts observed in other multi-task scenarios.
>
> Regarding the reference to "These contents can be shown in lines 259-256," I was unable to find the relevant content mentioned. Could the authors please clarify this?
>
> Author states that, “traditional multi-model merging scenarios mainly assume we have different task-specific models for merging, which only focus on the task conflict. But we not only consider the parameter conflict and but also consider the fine-grained alignment promotion simultaneously.” I would argue that alignment is, in itself, a form of task, and the different alignment outcomes are a result of parameter updates. Therefore, I still believe that the proposed method is not fundamentally different from those applied to other downstream tasks, aside from its specific focus on the 3H objectives.
>
> The results in Fig. 6, as well as the majority of results in Tab 3 & 4, appear to be within the margin of random fluctuation. The improvements in Tab 3 & 4 seem to be primarily driven by the OR-Bench's results. For a method like model merging, which inherently introduces uncertainty, such marginal improvements are insufficient to convincingly demonstrate the method's effectiveness.
>
>
> Given the unresolved issues above, I am inclined to maintain my original score

---

> ### Author Response · Authors · 2025-08-07
> **Reply to Reviewer HsEK**
>
> Thanks for your questions again! We are glad to address your concerns further.
>
> **Q1:The fundamental distinction between 3H alignment and other downstream tasks.**
>
> We would like to strengthen the distinction between 3H alignment and other downstream tasks as follows:
>
> **(1) Different Optimization Objectives**
>
> For the multi-task learning setting, it has clear optimization directions adapted to downstream tasks, but for 3H alignment, it requires **not just “task performance” but also ethical and societal acceptability.** Thus, it's usually defined by complex, human-centric preference signals (often noisy, subjective, and multi-faceted), rather than a clear objective(ground truth) for downstream tasks. In this way, **we can treat the helpful part of 3H alignment(different tasks) as a multi-task setting, but considering the 3H alignment of LLM training, it's more complex**, because we need to consider high-level Human Preference demand(ethical and societal acceptability) rather simply fitting in data from the same or different domains like previous merging works under the supervised fine-tuning training for different tasks.
>
> **(2) Different Inherent Meaning of Parameter Updates**
>
> Discussions about parameter updates of the multi-task merging mainly focus on the similarity analysis from the domain perspective. But the relationship among 3H dimensions is different from multi-task learning. Considering the process of preference data construction, **the harmlessness and honesty dimension are naturally embedded into the helpful training data. This means we should consider the ethical and societal acceptability based on existing tasks, while previous domain-based multi-model merging works don't have the strict dependence**. Thus, the parameter updates of 3H alignment belong to a more fine-grained and complex demand.
>
> **(3) New findings on existing multi-model merging methods for LLM 3H alignment**
>
>  **(i) Most of the multi-model merging methods achieve worse results than data mixture methods.** For example, in Table 3, only 3/13 previous merging methods achieve better results than data mixture strategies. However, our proposed RESM can consistently achieve results comparable to previous methods. **The reason is that we treat the multi-task learning and 3H alignment differently.**
>
> (ii) The advancement of multi-model merging methods has the same assumption of separating task-common and task-specific parameters to cope with conflict. For 3H alignment, the investigation analysis of Figure 6 and Broad Impact and Limitation shows that **LLM 3H alignment is a more complicated problem; we can't directly succeed this assumption, as the worse experimental results can support our analysis.** Thus, we adopt the layer-wise SVD composition to seek the orthogonalization for 3H merging.
>
> (iii) There exist special problems for LLM 3H merging, as stated by **Preference Noise Accumulation and Effective Rank Analysis, where we further discuss the drawbacks of previous multi-model merging methods.**
>
>
> **Q2: More clarification for Contribution.**
>
> **(1) Valuable Problem**
>
> Until now, **we are the first to systematically scale the data mixture and model merging strategies for LLM 3H optimization. Traditional multi-model merging methods don't deal with this comparison.** The detailed experimental results of LLMs across different human preferences can contribute to large-scale industrial applications compared with traditional multi-model merging methods.
>
> **(2) Reasonable Technique**
>
> We aim to provide insights to **(i) analyse the collaborative and conflict relationships of LLM 3H alignment**, **(ii) understand the theoretical mechanism​ of 3H model merging**, including when merging is better and the factors that influence the merging effect, and **(iii) quantify the difference between data mixture and model merging methods for LLM 3H alignment**. These conclusions can contribute to the community to have a better understanding of the merging and data mixture strategies for LLM alignment.
>
> **(3) Sufficient Experimental Results**
>
> The comparison results should be evaluated from **(i) the difference between data mixture methods and model merging methods** (lines267-285) (ii) **the effectiveness of different merging methods** (lines286-301) (iii) **static optimization and continue optimization to discuss the generalization of our methods** (lines302-313) (iii) **robustness of merging methods to discuss the uncertainty of mering methods as you said** in the paper (lines779-784). These experiments can verify the effectiveness of our method.
>
> We have tried our best to ensure a fair and reasonable training and evaluation in order to have a meaningful conclusion for LLM alignment. We hope these replies can address your questions, and we will add these discussions and analyses in the revised paper, referring to your reply. We sincerely hope you can re-evaluate our paper.

---

> ### Comment · Reviewer_HsEK · 2025-08-08
>
> I acknowledge the authors' rebuttal. But I find my primary concerns remain unanswered, and thus I will stand by my initial score.

---

> ### Author Response · Authors · 2025-08-09
> **Reply to Reviewer HsEK**
>
> Thanks for your reply! Could you provide more details about your concerns and provide an explanation of why "primary concerns remain unanswered"? We are glad to address them until the end of the rebuttal!
>
> >Q1: I am still not convinced about the fundamental distinction between 3H alignment and other downstream tasks
>
> Reply to Q1: The fundamental distinction between 3H alignment and multi-task learning has been clarified from different perspectives in the rebuttal.
>
> >Q2: For a method like model merging, which inherently introduces uncertainty, such marginal improvements are insufficient to convincingly demonstrate the method's effectiveness.
>
> Reply to Q2: Our sufficient experiments have considered the uncertainty you mentioned, and we have tried our best to ensure the reasonability of the experiments. Please see the above rebuttal for more details. More importantly, apart from experimental results, we also clarify our contribution from the perspective of the problem, technique, and theoretical insights for LLM alignment.
>
> If you have other concerns, we are glad to address them until the end of the rebuttal! We are grateful for your time and reviews to help us polish our paper. We will carefully organize our paper, based on your suggestions.

---

> ### Comment · Reviewer_qyir · 2025-08-09
> **Reply to Reviewer HsEK by Reviewer qyir**
>
> As a reviewer of this manuscript, I have carefully read your review comments. From the perspective of a peer reviewer, I find your feedback to be overly superficial and vague. Your questions could be more specific and concrete. Only by first learning to serve as a competent reviewer can one ensure fair evaluation of their own work.

---

### Official Review · Reviewer_qyir · 2025-07-03

**Clarity:** 3
**Significance:** 4
**Originality:** 3
**Rating:** 5
**Confidence:** 4

**Summary:**

This paper addresses the challenge of achieving balanced alignment of large language models (LLMs) across three critical dimensions—helpfulness, honesty, and harmlessness (3H optimization)—through model merging rather than traditional data mixture strategies. While existing methods face limitations such as reliance on expert knowledge and conflicting optimization signals, the authors propose a novel parameter-level conflict-resolution framework via model merging. Their approach, termed Reweighting-Enhanced Task Singular Merging (RESM), introduces outlier-aware weighting and sparsity-adaptive rank selection to mitigate preference noise accumulation and layer sparsity adaptation issues inherent in merging 3H-aligned LLMs. Theoretical analyses and experiments on benchmarks like Llama3 demonstrate RESM’s superiority over prior data mixture and model merging techniques, achieving notable performance gains in balanced alignment metrics.

The work further distinguishes itself by addressing gaps in training-free merging strategies, contrasting with methods like H3 fusion that require additional data for adaptation. By emphasizing reproducibility through detailed implementation guidelines and ablation studies, the authors validate RESM’s robustness in preserving task-relevant information during merging. They also contextualize their contributions within broader discussions of societal impacts, ethical considerations, and limitations, positioning RESM as a pathway for ethically constrained LLM development without compromising technical efficacy.

**Questions:**

1. The theoretical analysis assumes Gaussian-distributed preference noise and layer sparsity (e.g., outlier detection via Z-scores). However, real-world LLM training often exhibits heavy-tailed or non-stationary noise (e.g., power-law distributions of gradient updates). How robust is RESM to deviations from these assumptions?

2. How does RESM perform on sparse or modular architectures (e.g., mixture-of-experts models) compared to dense LLMs? The paper focuses on dense models like Llama3 and Mistral, but many modern LLMs use sparse designs to improve efficiency. Does RESM’s reliance on SVD-based sparsity adaptation and outlier-aware weighting remain effective when layer-specific parameter updates are inherently sparse or distributed across experts?

**Ethical Concerns:**

["NO or VERY MINOR ethics concerns only"]

**Final Justification:**

he authors have addressed my raised concerns, and I maintain my initial rating.

**Limitations:**

Yes.

**Paper Formatting Concerns:**

No.

**Quality:**

3

**Strengths And Weaknesses:**

**Strengths:**

1. The paper introduces RESM, a novel model-merging framework that explicitly addresses preference noise accumulation and layer sparsity adaptation in balancing helpfulness, honesty, and harmlessness. This fills a critical gap in existing methods, which often struggle with conflicting optimization signals or rely on expert knowledge. The combination of outlier-aware weighting and sparsity-adaptive rank selection provides a principled approach to parameter-level conflict resolution, advancing the field of ethically constrained LLM alignment.

2. By benchmarking model merging against traditional data mixture strategies, the work reveals previously overlooked collaborative and conflicting relationships among the 3H dimensions. This systematic comparison not only highlights the limitations of data-level approaches but also establishes a foundation for understanding how parameter-level strategies like RESM can achieve balanced alignment. Such insights are valuable for both practitioners and researchers aiming to navigate trade-offs in multi-objective LLM optimization.

3. The authors demonstrate RESM’s superiority through extensive experiments on benchmarks like Llama3 and Mistral, achieving measurable gains (2%-5% over data mixture, 1%-3% over prior merging methods). The inclusion of ablation studies, theoretical analyses (e.g., effective rank analysis), and reproducibility guidelines strengthens the credibility of their claims. This emphasis on empirical rigor ensures the method’s practical applicability and transparency.

---

**Weaknesses:**

1. The theoretical analysis assumes that preference noise and layer sparsity follow specific statistical distributions (e.g., Gaussian for outlier detection). However, real-world LLM training dynamics often involve heavy-tailed or non-stationary distributions, which could undermine RESM’s effectiveness. The paper does not empirically validate these assumptions, leaving gaps in understanding how robust the method is to deviations from idealized conditions.

2. Although RESM avoids data-dependent training, its implementation requires fine-grained control over parameter merging (e.g., rank selection, outlier thresholds), which may demand significant computational resources and domain expertise. The paper does not thoroughly discuss the trade-offs between merging complexity and alignment performance, such as how hyperparameter tuning scales with model size or task diversity. This could hinder adoption in resource-constrained settings or real-time applications.

---

> ### Author Rebuttal · Authors · 2025-07-31
>
> Thanks for your valuable suggestions! We are glad to address your concerns one by one.
>
> **Q1: Theoretical analysis assumes Gaussian-distributed preference noise and layer sparsity**
>
> Thanks for your questions. We acknowledge the outlier detection via Z-scores have the assumption of Gaussian-distributed and 3σ  may fail in a heavy-tailed distribution. But notably, for our model merging setting, the outlier have different meaning compared with traditional training.
>
> **(1) The outlier weighting is only operated on the singular vector of each alignment objective.**
>
> **(2) The outlier weighting is calculated based on the same layer of different models rather than different layers of one trained model**.
>
> **(3) This outlier noise mainly belong to the dominant factors that weaken multi-objective alignment of 3H merging rather than traditional noise for model training**.
>
>  Thus, we just assume that the same layer of different models have the Gaussian-distributed due to the few parameters undergo significant changes while most exhibit minor adjustments. The design of outlier weighting can conclude as follows: For noise compression, we filter out low-magnitude fluctuations that predominantly encode noise, forcing the singular vectors to align with statistically significant task features. ensuring balanced contributions from diverse tasks and preventing dominance by high-magnitude updates that may obscure subtle yet critical features. For task equilibrium: The layer-wise aggregation outlier weights are globally normalized across all models, ensuring balanced contributions from diverse tasks and preventing dominance by high-magnitude updates that may obscure subtle yet critical features. These contents can be shown in lines182-188.
>
> **Q2: The paper does not thoroughly discuss the trade-offs between merging complexity and alignment performance.**
>
> **(1) For rank selection(related to sparsity)**, we carefully discuss the sparsity and random seed that influences the effect of merging in Figure 5 and Figure 6, we can observe that RESM can achieve better robust results than previous random sparsification-based methods, further verifying the effectiveness of our methods. The experiments that can verify the robustness of our proposed RESM can be shown in Appendix C.4.
>
> **(2) For outlier detection**, we follow the 3σ principle to cope with the preference noise accumulation shown in Figure 2. It has a stable effect on the model merging.
>
> **(3) For task diversity**, as shown in Table 1, we have conducted experiments through 5 categories of alignment objectives across 10 different datasets. The reported results can verify the effectiveness of our method.
>
> **Q3: How does RESM perform on sparse or modular architectures (e.g., mixture-of-experts models) compared to dense LLMs? Does RESM’s reliance on SVD-based sparsity adaptation and outlier-aware weighting remain effective when layer-specific parameter updates are inherently sparse or distributed across experts?**
>
> Thanks for your questions. The operated object of SVD-based sparsity adaptation and outlier-aware weighting are layer-wise parameters of different models rather than different layer of one models. Thus, we believe our proposed RESM can still achieve good performance for 3H alignment. Due to the time limit, we will add the experiments later

---

> > ### Comment · Reviewer_qyir · 2025-08-06
> >
> > I appreciate the authors' response, and I maintain my initial rating.

---

> > > ### Author Response · Authors · 2025-08-07
> > > **Reply to Reviewer qyir**
> > >
> > > Thanks for your advice and question again. We will carefully polish our paper, referring to your reviews.

---

### Comment · Area_Chair_GUHX · 2025-08-04

Dear Authors and Reviewers,

I would like to thank the authors for providing detailed rebuttal messages

To reviewers: I would like to encourage you to carefully read all other reviews and the author responses and engage in an open exchange with the authors. Please post your first response as soon as possible within the discussion time window. Ideally, all reviewers will respond to the authors, so that the authors know their rebuttal has been read.

Best regards,
AC

---

### Decision · Program_Chairs · 2025-09-17

**Decision:**

Accept (poster)

**Comment:**

This paper provides a systematic comparison of data mixture versus model merging strategies for aligning Large Language Models (LLMs) with the principles of Helpfulness, Honesty, and Harmlessness (3H). The authors argue that while data-level mixture methods face challenges with conflicting optimization signals, parameter-level model merging offers a promising alternative. To address the specific issues of preference noise and layer sparsity in 3H alignment, the paper introduces a novel merging method called Reweighting-Enhanced task Singular Merging (RESM). Through extensive experiments, the authors demonstrate that RESM outperforms both existing data mixture and model merging techniques, establishing a strong benchmark for future work in multi-objective LLM alignment.

The paper's primary strength, as highlighted by a majority of the reviewers, is its pioneering role in systematically investigating and benchmarking data mixture against model merging for 3H alignment. This is a critical and timely problem for responsible AI, and the comprehensive benchmark itself is a valuable contribution to the community. The proposed RESM method is technically sound, offering a principled approach to mitigate specific challenges in merging 3H-aligned models, such as preference noise and layer sparsity. Finally, the work is supported by extensive and thorough experiments that validate the effectiveness of RESM across multiple models and datasets, showing clear performance gains over numerous baselines.

The main point of contention, raised forcefully by Reviewer HsEK, is whether the "3H conflict" is a fundamentally unique problem or simply another instance of standard multi-task parameter conflict. This questions the core motivation and significance of the work, suggesting that the insights might not be specific to the 3H scenario.

I recommend accepting this paper. While the concerns raised by Reviewer HsEK regarding the fundamental novelty of the problem are valid points for discussion, the majority of reviewers (qyir, jwRS, UavS) found the paper to be a significant and valuable contribution. The primary value of this work lies in being the first to provide a large-scale, systematic empirical study of model merging for the critical task of 3H alignment. This benchmark and the corresponding analysis provide important insights for the community, regardless of whether 3H conflicts are a completely new class of problem. The proposed RESM method is technically solid and demonstrably effective. Furthermore, the authors were diligent in their rebuttal, providing new experiments on more recent models that addressed the concerns of multiple reviewers and led Reviewer UavS to raise their score.